# Caspase-1 cleaves PPARγ for potentiating the pro-tumor action of TAMs

Zhiyuan Niu[1], Qian Shi[2], Wenlong Zhang[1], Yuxin Shu[1], Nanfei Yang[1], Bing Chen[3], Qingsong Wang[4], Xuyang Zhao[4], Jiajia Chen[1], Nan Cheng[1], Xiujing Feng[1], Zichun Hua[5], Jianguo Ji[4] & Pingping Shen[1]

Tumor-associated macrophages are increasingly viewed as a target of great relevance in the tumor microenvironment, because of their important role in cancer progression and metastasis. However, the endogenous regulatory mechanisms underlying tumor-associated macrophage differentiation remain largely unknown. Here, we report that caspase-1 promotes tumor-associated macrophage differentiation by cleaving peroxisome proliferator-activated receptor gamma (PPARγ) at Asp64, thus generating a 41 kDa fragment. This truncated PPARγ translocates to mitochondria, where it directly interacts with medium-chain acyl-CoA dehydrogenase (MCAD). This binding event attenuates MCAD activity and inhibits fatty acid oxidation, thereby leading to the accumulation of lipid droplets and promoting tumor-associated macrophage differentiation. Furthermore, the administration of caspase-1 inhibitors or the infusion of bone marrow-derived macrophages genetically engineered to overexpress murine MCAD markedly suppresses tumor growth. Therefore, targeting the caspase-1/PPARγ/MCAD pathway might be a promising therapeutic approach to prevent tumor progression.

---

[1] State Key Laboratory of Pharmaceutical Biotechnology and MOE Key Laboratory of Model Animal for Disease Study, Model Animal Research Center, Nanjing University, Nanjing 210023, China. [2] Department of Cellular and Integrative Physiology, The University of Texas Health Science Center at San Antonio, STRF-Greehey North Campus, 8403 Floyd Curl Drive, San Antonio, Texas 78229-3904, USA. [3] Department of Hematology, Nanjing Drum Tower Hospital, the Affiliated Hospital of Nanjing University Medical School, Nanjing 210008, China. [4] The State Key Laboratory of Protein and Plant Gene Research, College of Life Sciences, Peking University, Beijing 100871, China. [5] State Key Laboratory of Pharmaceutical Biotechnology School of Life Sciences, Nanjing University, Nanjing 210023, China. Zhiyuan Niu and Qian Shi contributed equally to this work. Correspondence and requests for materials should be addressed to P.S. (email: ppshen@nju.edu.cn)

Tumor-associated macrophages (TAMs) reside in the tumor microenvironment and are the primary components of leukocyte infiltrates. TAMs express factors that promote angiogenesis and tissue remodeling and inhibit the anti-tumor immune response in breast, prostate, cervical, and ovarian cancers. High levels of TAMs in these cancers are correlated with poor prognoses[1]. In response to microenvironmental signals, inactivated macrophages (M0) differentiate into M1 (classically activated) like, M2 (alternatively activated) like macrophages or other unknown polarized macrophages; It is now generally accepted that the term TAMs actually describes the macrophages infiltrating in the tumor environment not displaying the same phenotype with classical M1 or M2 macrophages, which depends on the cytokine balance of the tumor microenvironment, remaining to be illustrated comprehensively[2,3]. This inflammation activates the tumorigenic functions of TAMs, such as those associated with tumor cell survival, proliferation, and dissemination[4]. Given the important roles of TAMs in tumor

development, targeting TAM functions such as cell recruitment[5], survival[6] and polarization[7] has recently emerged as a novel therapeutic approach to inhibit cancer progression. However, the therapeutic application of TAMs is still in its infancy, and TAM-associated therapeutic strategies have provided only modest clinical benefits. Therefore, we sought to investigate new methods to modulate the tumorigenic functions of TAMs.

Metabolic adaptation is a key feature of macrophage plasticity and polarization and is instrumental to macrophage function in homeostasis, immunity, and inflammation[8]. For example, arginine metabolism is a feature of macrophage polarization that is characterized by high levels of inducible nitric oxide synthase (iNOS) in M1 macrophages and high levels of arginase in M2 polarized macrophages[9]. Differential metabolic programs are essential for the function of cells in the innate and adaptive immune system[10]. The metabolism of M1 macrophages is characterized by increased glycolytic flux and reduced mitochondrial oxidative phosphorylation, as compared with

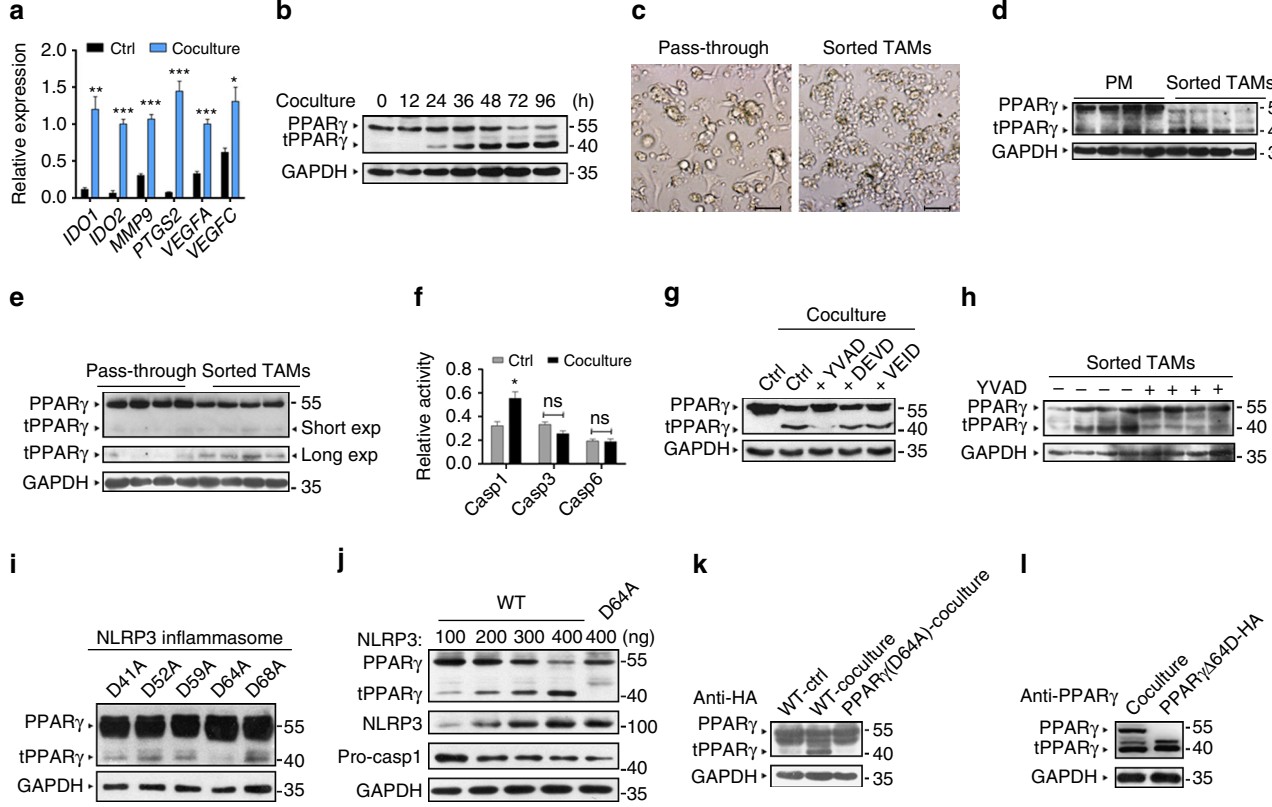

**Fig. 1** Caspase-1 mediates PPARγ cleavage in TAMs in vitro and in vivo. **a** THP-1 macrophages were cultured alone or cocultured with MCF-7 for 48 h, the expression of protumoral genes was assessed by RT-qPCR (n = 6). **b** Cocultured THP-1 cells were harvested at indicated time, PPARγ and its truncated fragment (tPPARγ) were determined by immunoblot. The expression levels of proteins in this figure were all determined by immunoblot, except **a**, **c** and **f**. **c** Cell morphology of sorted TAMs and pass-through cells (mainly tumor cells) from MMTV-PyMT mice. Scale bars, 100 μm. **d**, **e** PPARγ and tPPARγ in TAMs from MMTV-PyMT mice (**d**), or 4T1 orthotopic-grafted mice (**e**) were shown, using peritoneal macrophages (PM) (**d**) or pass-through cells (**e**) as the control. TAMs in each different lane sample was pooled from at least two mice (the same in **i**). **f** THP-1 macrophages were cultured alone or cocultured with MCF-7 for 24 h, then the relative activity of caspase-1, caspase-3, or caspase-6 was tested (n = 3). All statistic data represent means ± s.e. m., (*P < 0.05, ** < 0.01, *** < 0.001, ns, not significant, on way ANOVA for multiple comparisons). **g** THP-1 macrophages were cultured alone or cocultured with MCF-7 in the presence or absence of caspase inhibitors for 48 h, PPARγ and tPPARγ were shown here. **h** PPARγ and tPPARγ in sorted TAMs from MMTV-PyMT mice administration of vehicle or caspase-1 inhibitor YVAD, were determined by immunoblot. **i**, **j** The levels of PPARγ and tPPARγ were shown in HEK293T cells (**i**), which were transfected with different PPARγ mutant plasmids as indicated and inflammasome associated plasmids (NLRP3 300 ng, pro-caspase-1 20 ng and ASC 20 ng) for 24 h (**i**), or HEK293T cells (**j**), which were transfected with PPARγ or PPARγ (D64A) plasmids together with plasmids for inflammasome with increasing doses for NLRP3 (100 to 400 ng). **k** Extracts of THP-1 cells transfected with HA-PPARγ plasmids were cultured alone or cocultured with MCF-7 for 48 h and extracts of THP-1 cells transfected with HA-PPARγ (D64A) plasmids cocultured with MCF-7 for 48 h were determined by immunoblot. **l** PPARγ in extracts of THP-1 macrophages cocultured with MCF-7 for 48 h, and HEK293T cells transfected with HA-PPARγΔ64D plasmids were shown

M0 cells. In contrast, oxidative glucose metabolism (fatty acid oxidation) is the metabolic pathway favored by IL-4-activated M2 macrophages[11]. Emerging data support the concept that metabolic regulators control macrophage activation. Inhibition of fatty acid oxidation dramatically decreases the activation of M2 but not M1, macrophages[12]. Selective reprogramming of fatty acid metabolism alters the inflammatory response in macrophages[13], and glutamine deprivation or inhibition of N-glycosylation decreases M2-prone polarization. Consistently with these findings, inhibition of aspartate-amino transferase suppresses nitric oxide and interleukin-6 production in M1-like macrophages[14]. Moreover, the modulation of macrophage function has emerged as an off-target effect of PPARγ agonists[15] and statins, which are well-characterized metabolic regulators.

The metabolic states of both M1 and M2 macrophages have been well characterized. Classically activated macrophages (M1) preferentially utilize glucose, whereas alternatively activated macrophages (M2) utilize fatty acid oxidation for energy homeostasis[12]. However, the metabolic profiles and the mechanisms underlying TAM metabolism remain poorly characterized. Therefore, detailed studies are required to delineate the metabolic profiles in TAMs and to determine how these distinct metabolic pathways are integrated into a coherent program that directs macrophage gene expression in TAMs. Such studies are necessary to support further investigation into the clinical applications of targeting TAMs in tumor immunotherapy. In this study, we found that caspase-1 inactivates MCAD by cleaving PPARγ and induces lipid accumulation TAMs. MCAD activity and lipid levels were strongly associated with TAM differentiation. Finally, modulating MCAD activity in TAMs through treatment with caspase-1 inhibitors or genetic manipulation suppressed primary tumor growth in in vivo mouse models, thus supporting potential clinical applications for targeting the caspase-1/PPARγ/MCAD pathway in TAM differentiation.

## Results

**Caspase-1 mediates PPARγ cleavage in TAMs.** The human monocytic leukemia cell line THP-1, after stimulation with phorbol-12-myristate 13-acetate (PMA) for 24 h, shares many properties with human monocyte-derived macrophages[16]. Here we examined the ability of PMA-treated THP-1 macrophages (referred to as THP-1 macrophages) to promote metastasis, invasion and angiogenesis in response to coculture with the human breast cancer cell line MCF-7 in separate chambers. The coculture model is presented in Supplementary Fig. 1. An analysis of the gene expression profile of cocultured THP-1 cells demonstrated elevated levels of genes that mediate T cell inhibition (*IDO1,2*), angiogenesis (*PTGS2*, *VEGFA*, *VEGFC*) and metastasis (*MMP9*; Fig. 1a). The functions of these genes are indicative of an enhancement in the tumorigenic functions associated with TAMs. These data suggested that THP-1 macrophages acquired a TAM-like phenotype in this coculture system.

Notably, PPARγ cleavage was observed in THP-1 cells cocultured with MCF-7 cells for 24 h (Fig. 1b). PPARγ cleavage resulted in the generation of a 41 kDa truncated product that was detected by an antibody recognizing the C-terminus of PPARγ. Next, we investigated whether PPARγ cleavage occurred in vivo. TAMs were isolated using magnetic cell sorting (MACS) microbeads, and the purity of TAMs was examined through microscopy (Fig. 1c) and quantified by flow cytometry (Supplementary Fig. 2a). Similar to the results of the coculture analysis, PPARγ cleavage was observed in primary TAMs isolated from

2 separate tumor models: a transgenic mouse model exhibiting spontaneous breast tumors-MMTV-PyMT mice model (Fig. 1d) and the orthotopic-grafted breast cancer mouse model (Fig. 1e) generated by injecting 4T1 cells into female 9-week-old BALB/c mice. To further verify the cleavage of PPARγ observed in the coculture model, nuclear fractions of THP-1 macrophages were prepared at the indicated time points and analyzed with a gel electrophoresis mobility shift assay (EMSA) with a PPARγ specific DNA probe. Surprisingly, as full-length PPARγ levels decreased, levels of the truncated product increased (Supplementary Fig. 2b).

Previous reports have demonstrated that PPARγ is cleaved by caspase-1, 3 and 6[17, 18]. We have found that PPARγ cleavage in cocultured TAMs is blocked by the pan-caspase inhibitor NCX-4016[19] (Supplementary Fig. 2c). Unexpectedly, we detected an increase in the activity of caspase-1, but not caspase-3 and caspase-6, in the coculture model (Fig. 1f). The activation of caspase-1 in TAMs in vivo was further confirmed by western blotting, as demonstrated by the appearance of the 20 kDa (p20) band (Supplementary Fig. 2d, e), and ELISA assay for the caspase-1 substrates IL-1β and IL-18 (Supplementary Fig. 2f). In cocultured TAMs, incubation with the caspase-1 inhibitor YVAD, but not caspase-3 inhibitor (DEVD) and caspase-6 inhibitor (VEID), blocked the generation of the 41 kDa PPARγ fragment (Fig. 1g), since specific inhibitors of caspase-3 and caspase-6 attenuate the cleavage of PPARγ protein in response to TNFα in cultured adipocytes[17, 18] the positive controls for inhibitors on caspase-3 and caspase-6 are shown in Supplementary Fig. 2g. Similarly, in primary TAMs derived from MMTV-PyMT mouse tumor tissues, the caspase-1 inhibitor YVAD inhibited PPARγ cleavage (Fig. 1h). A previous report has demonstrated that NLRP3 (NACHT, LRR and PYD domains-containing protein 3) inflammasomes efficiently trigger caspase-1 activation in macrophages[20]. To confirm that caspase-1 mediated the cleavage of PPARγ in TAMs and to identify the cleavage site(s) in PPARγ, we employed NLRP3 inflammasome reconstitution assay in HEK293T cells. We generated a panel of PPARγ mutants with Asp (D) to Ala (A) mutations in the putative cleavage sites, because aspartate is the signature residue of cysteine protease substrates. We found that only the 293-T cells transfected with the PPARγ (D64A) mutant in combination with inflammasome-associated plasmids failed to produce the 41 kDa cleavage product (Fig. 1i). Also, overexpression of HA-NLRP3 in 293-T cells significantly increased the activity of caspase-1 (decreased the expression of pro-caspase-1) and concomitantly increased the levels of the truncated form of wild type PPARγ but not PPARγ (D64A; Fig. 1j), thus confirming that PPARγ was cleaved by caspase-1 at Asp64. In addition, THP-1 cells were transfected with plasmids encoding the C-terminal HA-tagged wild type PPARγ or PPARγ (D64A) mutant and cocultured with MCF-7 cells for 48 h. The 41 kDa cleavage product was observed in cells transfected with wild-type PPARγ but was absent in cells transfected with the mutant PPARγ (D64A; Fig. 1k). Moreover, the putative site (PVVADYKYDL) is highly conserved between humans and mice (Supplementary Fig. 2h). To further confirm that the truncated PPARγ fragment (tPPARγ) was generated by cleavage at Asp64, HEK293T cells were transfected with plasmids expressing a C-terminal HA-tagged tPPARγΔ64D mutant with aa 1-64 deleted. HEK293T cells expressing tPPARγΔ64D did yield a fragment that appears at the corresponding region of the native tPPARγ fragment in cocultured THP-1 cells (Fig. 1l). The anti-PPARγ-C-terminal antibody sc-7273 recognized tPPARγ in cocultured THP-1 cells, but the anti-PPARγ-N-terminal antibody sc-7196 did not (Supplementary Fig. 2i, j). These findings further confirmed that caspase-1 cleaved the N-terminal of PPARγ during TAM differentiation.

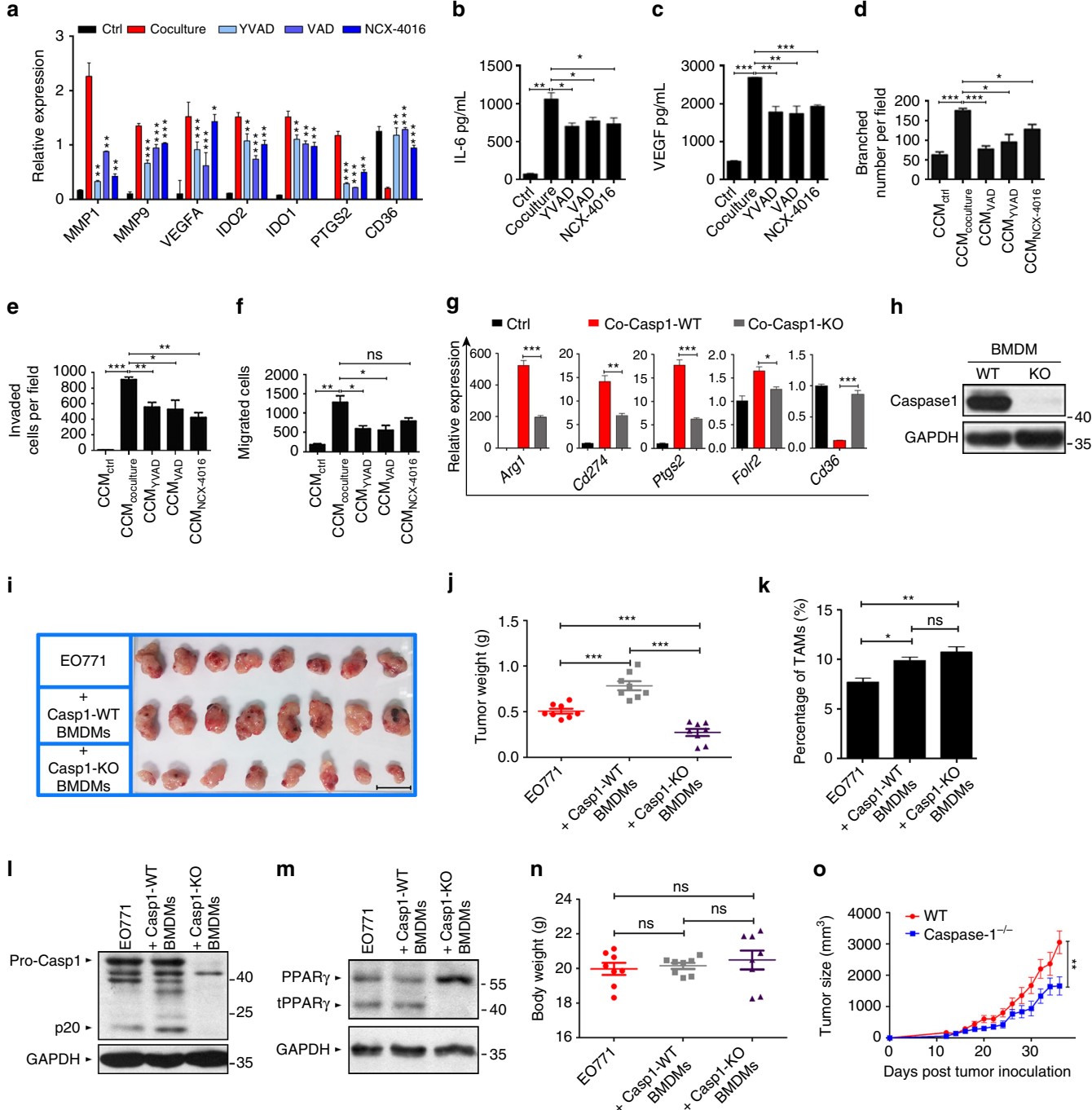

**Fig. 2** Caspase-1 inhibition prevents PPARγ cleavage and then blocks TAM differentiation. **a** THP-1 cells were cultured alone or cocultured with MCF-7 or cocultured in the presence of YVAD, VAD or NCX-4016 for 48 h, and the expression of pro-tumoral genes and *CD36* were assessed by RT-qPCR ($n = 3$). **b**, **c** The levels of VEGF and IL-6 in concentrated cell culture supernatant were measured by ELISA ($n = 3$). **d–f** The pro-angiogenesis, pro-invasion and pro-migration ability of CCM$_{Ctrl}$, CCM$_{Coculture}$, CCM$_{YVAD}$, CCM$_{VAD}$ and CCM$_{NCX-4016}$ was tested ($n = 3$). **g** Peritoneal macrophages were isolated from wild type (Co-Casp1-WT) or *caspase-1*$^{-/-}$ mice (Co-Casp1-KO) and cocultured with 4T1. After 2 days, pro-tumoral genes and *Cd36* were determined by RT-qPCR, using wild type peritoneal macrophages cultured alone as control (Ctrl). Data were generated and analyzed by Step one plus Real-Time PCR-system (AB Applied biosystem) ($n = 3$). **h** Caspase-1 expression in BMDMs from wild type or *caspase-1*$^{-/-}$ mice was determined by immunoblot. **i**, **j** EO771 cancer cells were implanted in the flank of nude mice alone or in combination with WT or caspase-1-deficient (KO) BMDMs. Tumors were resected and measured 21 days later (in j, $n = 8$). Scale bar, 2 cm. **k** TAMs were sorted from the total tumor tissue cells by magnetic cell sorting using MACS, and TAM percentage in mixed cells was analyzed ($n = 8$). **l** Caspase-1 expression in TAMs was analyzed by immunoblot. **m** PPARγ and its fragment in TAMs were analyzed by immunoblot. **n** Nude mice body weight was analyzed ($n = 8$). **o** Tumor growth was monitored by measuring volumes every 2 days with a caliber rule. Tumor volumes were calculated by the formula, $1/2(a \times b^2)$, where a is the long diameter and b is the short diameter (in mm; $n = 8$). All statistic data in this figure represent means ± s.e.m., (*$P < 0.05$, **$P < 0.01$, ***$P < 0.001$, ns, not significant, one way ANOVA for multiple comparisons)

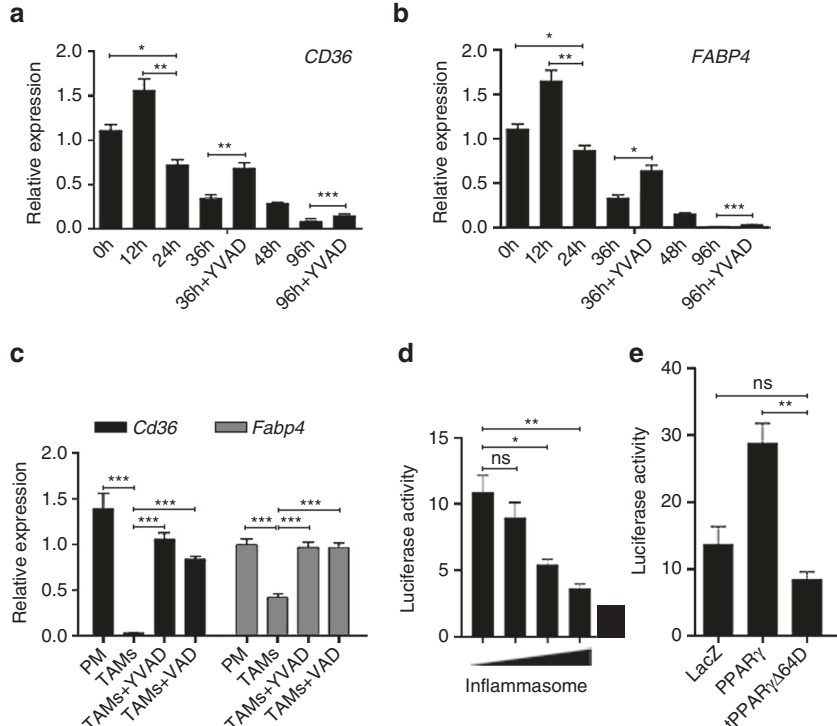

**Fig. 3** Caspase-1 cleavage attenuates the transcriptional activity of PPARγ. **a**, **b** THP-1 macrophages were cocultured with MCF-7 for indicated time, in the presence or absence of caspase-1 inhibitor YVAD. *CD36* and *FABP4* expression was assessed by RT-qPCR($n = 3$). **c** Peritoneal macrophages were isolated from tumor bearing MMTV-PyMT mice, and TAMs were isolated from tumor bearing MMTV-PyMT mice with administration of vehicle, YVAD or VAD, and then the expression of *Cd36* and *Fabp4* were assessed by RT-qPCR ($n = 8$). **d** HEK293T cells were transfected with PPARγ-luc reporter plasmids and PPARγ plasmids, along with increasing concentrations of NLRP3 (100, 200, 300, and 400 ng) and caspase-1, ASC plasmids, and then PPARγ-dependent luciferase activity was analyzed in cell lysates ($n = 3$). **e** HEK293T cells were transfected with PPARγ-luc reporter plasmids and plasmids expressing LacZ or HA-tagged PPARγ (PPARγ) or N-terminal truncated HA-tagged PPARγ at Asp64 (tPPARγΔ64D), and then the transcriptional activity of different PPARγ variants were analyzed ($n = 8$). All statistic data in this figure represent means ± s.e.m., (*$P < 0.05$, **$P < 0.01$, ***$P < 0.001$, ns, not significant, one way ANOVA for multiple comparisons)

**Inhibiting caspase-1 prevents PPARγ cleavage and TAM differentiation**. From our observations in cocultured THP-1 cells, we hypothesized that caspase-1-induced PPARγ cleavage plays a role in the generation of TAM-like cells. To evaluate this hypothesis, we analyzed the time course of expression of tumorigenic genes by using RT-qPCR. After 24 h in the coculture system, *IDO1*, *IDO2*, *PTGS2* were up-regulated in THP-1 cells (Supplementary Fig. 3a–c), at the time when PPARγ cleavage was initially observed. *MMP7*, *MMP9*, and *VEGFA* were also up-regulated but at later time points (48-96 h, Supplementary Fig. 3d–f). The changes in the protein levels of IDO1, MMP2, and MMP9 were consistent with the increased gene expression observed in the RT-qPCR analysis (Supplementary Fig. 3g–i). Then, we determined whether the up-regulation of tumorigenic genes was associated with caspase-1 activation. Indeed, multiple caspase inhibitors, including caspase-1 inhibitor YVAD, and the pan-caspase inhibitors VAD and NCX-4016 suppressed the up-regulation of tumorigenic genes in cocultured THP-1 cells (Fig. 2a).

In addition, we used the coculture model to mimic the local tumor microenvironment[21], by using 10 × concentrated conditioned medium (CCM) from THP-1 macrophages cultured alone (CCM$_{Ctrl}$), cocultured with MCF-7 (CCM$_{Coculture}$), or cocultured in the presence of the caspase inhibitors YVAD, VADs and NCX-4016 (referred to as CCM$_{YVAD}$, CCM$_{VAD}$ and CCM$_{NCX-4016}$, respectively) (procedural details in Methods and Supplementary Fig. 1). The cytokines VEGF and IL-6 were highly expressed in CCM$_{Coculture}$ compared with CCM$_{Ctrl}$ cells, and were

significantly decreased in cells cocultured with caspase-1 inhibitors (Fig. 2b, c). Then the pro-angiogenesis (Fig. 2d and Supplementary Fig. 4a), pro-invasion (Fig. 2e and Supplementary Fig. 4b) and pro-migration (Fig. 2f and Supplementary Fig. 4c) ability of CCM$_{Ctrl}$, CCM$_{Coculture}$, CCM$_{YVAD}$, CCM$_{VAD}$, and CCM$_{NCX-4016}$ was evaluated. The tumorigenic functions of CCM$_{Coculture}$ cells were markedly enhanced, as compared with CCM$_{Ctrl}$ cells, and the administration of caspase-1 inhibitors significantly attenuated this effect. Moreover, to rule out the possibility that treatment with inhibitors directly impacts tumor cells, caspase-1 knockdown THP-1 cells were constructed, and these experiments were repeated and the similar results were observed (Supplementary Fig. 5).

Then, we examined the expression of TAM hallmarks in peritoneal macrophages derived from *caspase-1*$^{-/-}$ mice, cocultured with the mouse mammary tumor cell line 4T1. Compared with cells from wild-type mice, macrophages derived from *caspase-1*$^{-/-}$ mice exhibited a significant reduction in the expression of TAM-associated genes, further suggesting that caspase-1 expression is closely associated with TAM differentiation (Fig. 2g). Caspase-1 expression in *caspase-1*$^{-/-}$ mice and wild-type mice was verified by western blotting (Fig. 2h). To Further confirm the detrimental role of TAM-intrinsic caspase-1 in vivo, EO771 cancer cells were implanted in the flank of nude mice alone or in combination with WT or caspase-1-deficient (KO) BMDMs derived from *caspase-1*$^{-/-}$ mice. Implantation of malignant cells admixed with WT BMDMs markedly promoted the tumor growth, whereas

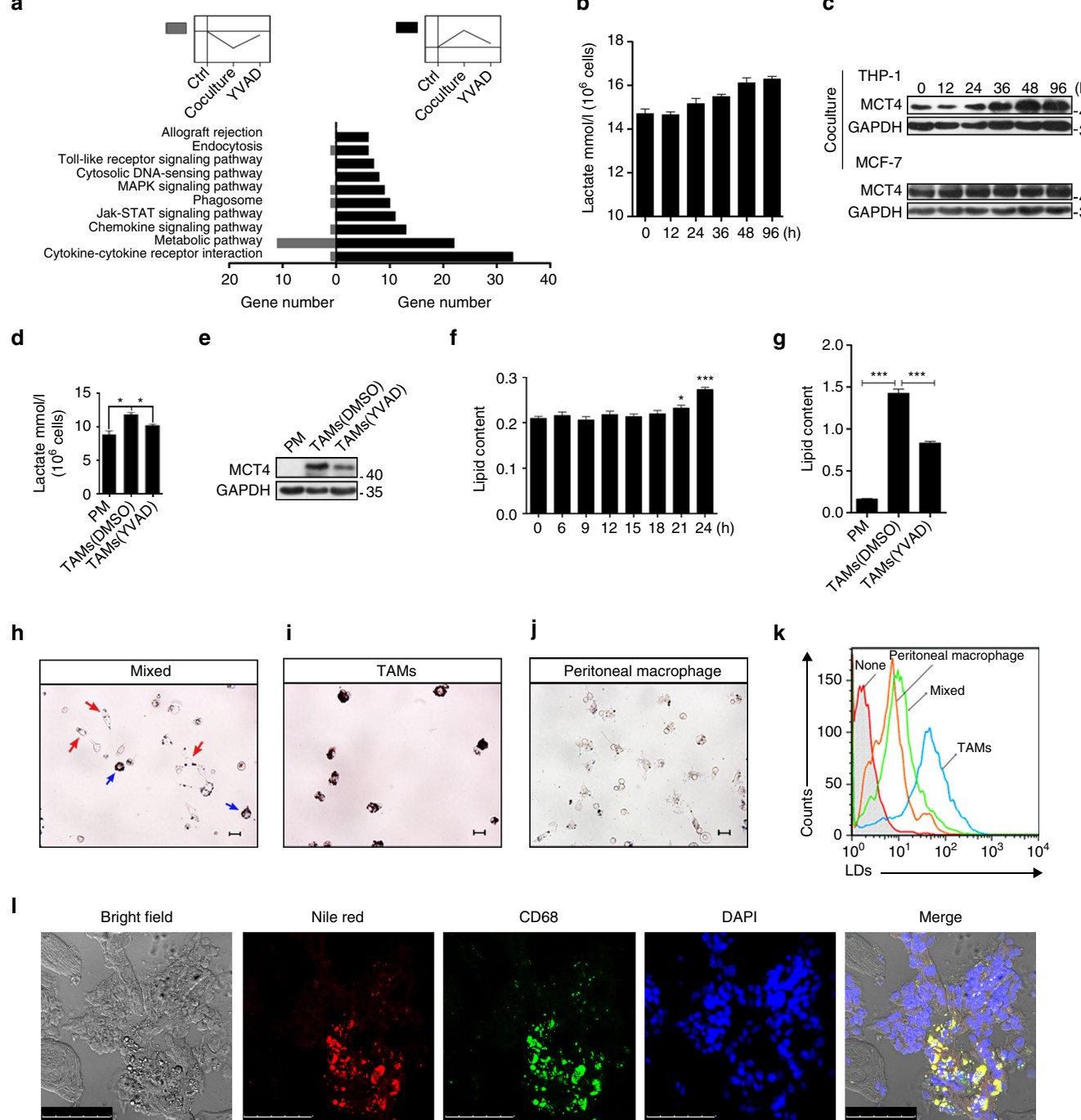

**Fig. 4** PPARγ cleavage causes a metabolic reprogramming in TAMs. **a** Pathway analysis was conducted on these 618 pre-selected genes. **b** Lactate secretion ability was measured ($n = 3$). **c** Protein level of MCT4 was analyzed by immunoblot, THP-1 (upper), MCF-7 (lower). **d**, **e** Peritoneal macrophages and TAMs were isolated from tumor-bearing MMTV-PyMT mice, MCT4 expression was analyzed by immunoblot (**e**). Or lactate concentration in the supernatant was quantified one day later (**d**; $n = 8$). **f** After cocultured for indicated time, THP-1 macrophages were stained by oil red, then analyzed lipid content by a quantitative method ($n = 3$). **g** Lipid content in peritoneal macrophages and TAMs from tumor bearing MMTV-PyMT mice was quantified ($n = 8$). **h** Single-cell suspensions from tumors were prepared and allowed to attach for 6 h, lipid content was shown by Oil Red O staining, tumor cells were marked by red arrows, lipid overloaded cells were marked by blue arrows. Scale bars, 10 μm. **i**, **j** Staining of TAMs (**i**) peritoneal macrophages (**j**) from MMTV-PyMT tumor-bearing mice with oil red. Scale bars, 10 μm. **k** Flow cytometry analysis of lipid levels in TAMs, tumor tissue whole population cells and peritoneal macrophage from MMTV-PyMT tumor-bearing mice with Nile Red staining. **l** The CD68-labeled TAMs and Nile Red-labeled lipid droplets were observed in frozen sections (from six patients). Scale bars, 75 μm. All statistic data in this figure represent means ± s.e.m., (\*$P < 0.05$, \*\*$P < 0.01$, \*\*\*$P < 0.001$, ns, not significant, one way ANOVA for multiple comparisons)

caspase-1-deficient BMDMs significantly impaired the tumor growth (Fig. 2i, j). We also observed slight changes of TAMs content in tumors (Fig. 2k), however, these changes seemingly do not correspond with tumor growth. To confirm the role of caspase-1 in promoting tumor growth, we analyzed caspase-1 (Fig. 2l) and truncated PPARγ fragment (Fig. 2m) in TAMs in each group, which validate the phenotype of BMDMs in this experiments. Same importantly here, body weight (Fig. 2n) remained the same between controls and experimental groups. In addition, EO771 cancer cells were orthotopically implanted in WT or $caspase-1^{-/-}$ mice, tumor growth was significantly impaired in $caspase-1^{-/-}$ mice (Fig. 2o). Together, these results demonstrate that caspase-1 expression in TAMs is necessary for the accelerated and aggressive progression of tumors.

Caspase-1 mediates the transition of the immature pro-inflammatory cytokines pro-IL-1β and pro-IL-18 to their mature and active forms. Accordingly, we observed that IL-1β and IL-18 were secreted in cocultured cells (Supplementary Fig. 6a, b). We wondered whether these cytokines were engaged by caspase-1 during TAM differentiation. Unexpectedly, exogenous administration of 600 pg ml$^{-1}$ IL-1β or 600 pg ml$^{-1}$ IL-18 (concentrations similar to those observed in the coculture system) did not induce any detectable changes (Supplementary Fig. 6c). These data implied that caspase-1 plays a new, previously uncharacterized role in regulating TAM differentiation.

To further evaluate the effect of the caspase-1 inhibitor YVAD on TAM differentiation, we used a microarray platform to comprehensively analyze gene expression in THP-1 macrophages cocultured with MCF-7 cells in the presence or absence of YVAD and in THP-1 macrophages alone. We specifically selected genes associated with caspase-1 cleavage, with a particular emphasis on genes whose expression patterns changed after treatment with caspase-1 inhibitors, for example, genes up-regulated in the coculture group (Coculture vs Ctrl) but down-regulated upon administration of YVAD (YVAD vs Coculture), or vice versa. Using a cutoff threshold of a 1.5-fold change in expression and a false discovery rate threshold of 0.05, we selected 618 genes for analysis (Supplementary Data 1). We found that genes that are typically upregulated in TAMs, including IDOs, PTGS2, MMPs and VEGF exhibited a similar trend. A gene ontology analysis[22] of the 618 genes revealed that 51 genes were strongly associated with the tumor-promoting functions of macrophages (Supplementary Fig. 7). Thus, the results of this microarray analysis indicated that the use of caspase-1 inhibitors might be an effective approach to regulate the tumorigenic functions of TAMs.

It has been well accepted that TAMs in cancer tissues bear complicated characteristics rather than simply M2 or M1, and, it is noteworthy that TAMs exhibit non classical M2/M1 phenotypes[2, 3] To answer the question whether caspase-1 mediated PPARγ cleavage plays an exclusive role in TAM differentiation, we investigated whether PPARγ was cleaved in macrophages exposed to stimulus of either classical activation (M1) or alternative activation (M2). The data suggest that PPARγ was selectively cleaved in the maturation of TAMs rather than in the polarization of either M1 or M2 (Supplementary Fig. 8a, b). The polarization of M1 or M2 was confirmed by nitric oxide synthase activity or arginase activity[23] (Supplementary Fig. 8c, d). We also compared typical M1 and M2 markers in undifferentiated primary macrophages and TAMs, the gene profiles of TAMs display a hybrid phenotype of M1 and M2 cells. In agreement with above evidence, we found that YVAD which potently inhibited TAM markers has little effect on M1 and M2 marker genes in TAMs (Supplementary Fig. 8e, f). Therefore, we conclude here that TAMs and M1, M2 share different differentiation mechanisms by which caspase-1 mediated PPARγ cleavage plays an exclusive role in the differentiation of TAMs.

**Caspase-1 cleavage impairs the transcriptional activity of PPARγ.** To explore the functional significance of PPARγ cleavage, we examined the activity of PPARγ in the coculture system. A time course study demonstrated that the mRNA levels of 2 classic PPARγ targets (FABP4, CD36) significantly decreased at 24 h, concurrently with PPARγ cleavage. As expected, the administration of the caspase-1 inhibitor YVAD to the coculture system for 36 and 96 h partially rescued the reduction in FABP4 and CD36 gene expression (Fig. 3a, b), we also showed that VAD and NCX-4016 or knockout of caspase-1 enhanced PPARγ activity in coculture (Fig. 2a, g), suggesting that caspase-1 activity was inversely related to PPARγ transcriptional activity. To determine whether a similar correlation exists in vivo, we compared Fabp4 and Cd36 levels in TAMs isolated from MMTV-PyMT tumor-bearing mice with peritoneal macrophages isolated from the same mice. As expected, levels of both Fabp4 and Cd36 significantly decreased in TAMs compared with peritoneal macrophages, and both YVAD and VAD blocked this effect. These findings indicated that PPARγ activity was significantly impaired in TAMs, and PPARγ activity can be enhanced by inhibiting caspase-1 (Fig. 3c). Additionally, we conducted NLRP3 inflammasome reconstitution assays in HEK293T cells by cotransfecting cells with plasmids expressing NLRP3, ASC or pro-caspase-1 with the PPARγ-luc reporter plasmid and analyzing PPARγ-dependent luciferase activity. We found that the activity of PPARγ strongly decreased with increasing levels of the NLPR3 expression vector that triggers caspase-1 activity (Fig. 3d). These results further confirmed the observation that caspase-1 attenuated PPARγ activity during TAM's maturation.

The decreased transcriptional activity of tPPARγ was probably due to the loss of its N-terminus because the expression of tPPARγΔ64D significantly reduced PPARγ-luciferase activity in 293-T cells compared with wild type PPARγ (Fig. 3e).

**PPARγ cleavage induces metabolic reprogramming in TAMs.** From our previous findings, we speculated that caspase-1 cleaves PPARγ and impairs its transcriptional activity, thereby promoting TAM differentiation. To determine the precise mechanism underlying the role of caspase-1 in TAM differentiation, we further analyzed multiple genes associated with PPARγ cleavage identified in the microarray analysis. First, multiple genes related to the 'cleavage incident' as previously described were selected. To evaluate whether pathway classification analysis might aid in identifying the biological pathways most likely to be involved in TAM differentiation, we performed pathway classification analysis on the 618 pre-selected genes. The 10 most highly represented pathways were associated with cytokine-cytokine receptor interactions, metabolism, chemokine signaling, Jak-STAT signaling, phagosomes, MAPK signaling, cytosolic DNA-sensing, Toll-like receptor signaling, endocytosis and allograft rejection (Fig. 4a and Supplementary Data 2). Genes associated with cytokine-cytokine receptor interactions and metabolic pathways were the 2 most highly represented pathways, with more than 30 genes represented in these 2 groups. These observations led us to speculate that metabolic reprogramming might be associated with caspase-1/PPARγ-mediated TAM differentiation. Next, we conducted Gene Ontology analysis and Pathway analysis of the differentially expressed genes associated with metabolism in TAM-like cells. We found that genes that function in lipid metabolism and carbohydrate metabolism were differentially expressed in cocultured cells compared with the control cells (Supplementary Fig. 9a, b and Supplementary Data 3, 4).

To further investigate whether metabolic reprogramming is involved in TAM differentiation, we evaluated metabolic alteration of TAMs on the basis of two indicators: lactate secretion and LD (lipid droplet) content. Lactate secretion significantly

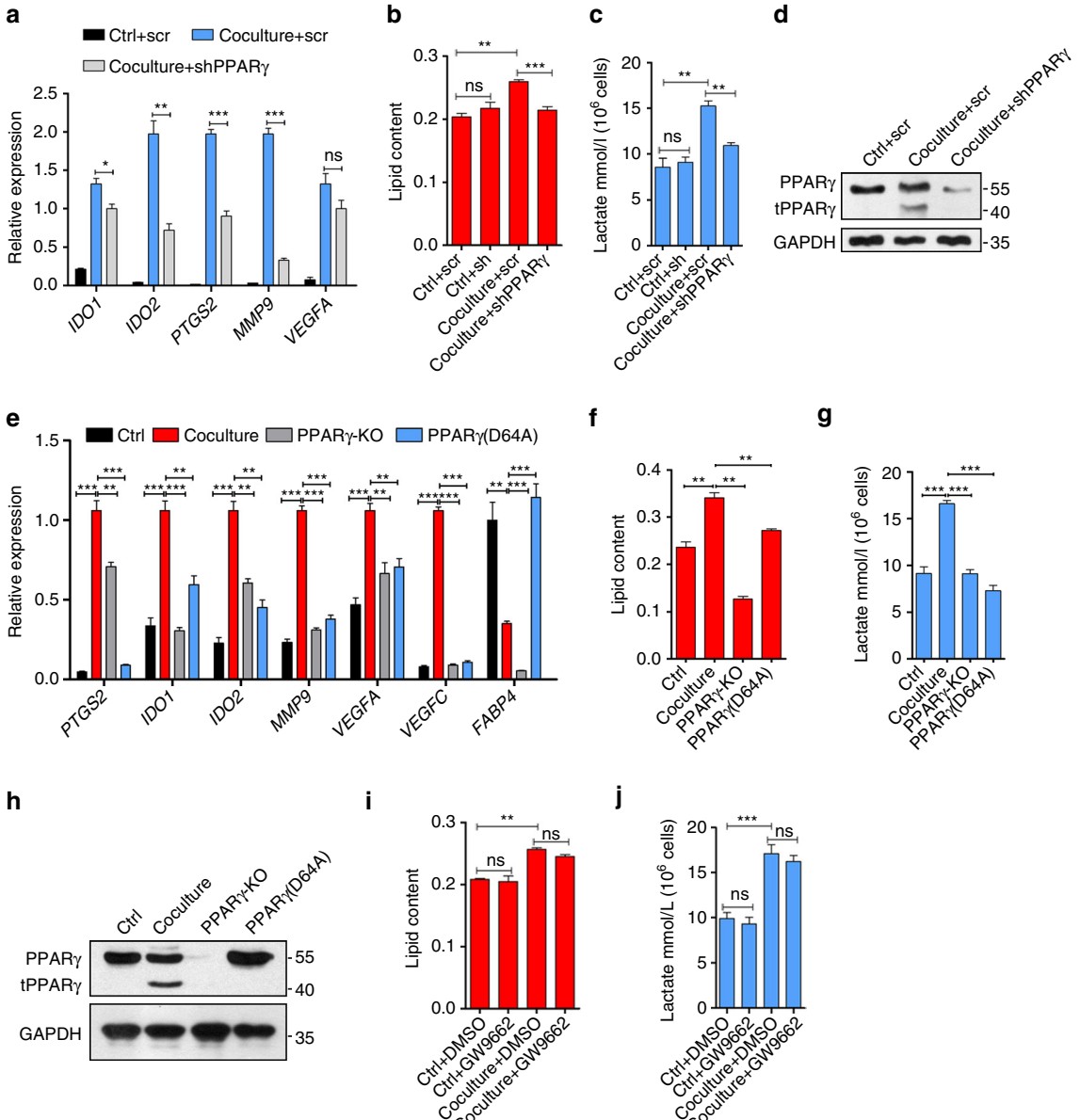

**Fig. 5** Loss of function of PPARγ fails to promote TAM differentiation or metabolic reprogramming in TAMs. **a–d** THP-1 cells stably expressing control scramble shRNA were cultured alone (Ctrl + scr) or cocultured with MCF-7 (Coculture + scr), THP-1 cells stably expressing shRNA against PPARγ were cultured alone (Ctrl + shPPARγ) or cocultured with MCF-7 (Coculture + shPPARγ), after 2 d, TAM hallmarks expression was analyzed by RT-qPCR (**a**), THP-1 macrophages were stained with Oil Red O and lipid content was quantified (**b**), or MCF-7 and regular culture medium was replaced with 2 ml fresh culture medium then culture for another 24 h, the lactate concentration were measured (**c**), PPARγ and its fragment were analyzed by immunoblot (**d**). **e–h** THP-1 cells stably expressing LacZ were cultured alone or cocultured with MCF-7, PPARγ knockout THP-1 cells or THP-1 cells that express PPARγ (D64A) were cocultured with MCF-7, after 2 d, TAM hallmarks expression was analyzed by RT-qPCR (**e**), THP-1 macrophages were stained with Oil Red O and lipid content was quantified (**f**), or MCF-7 and regular culture medium was replaced with 2 ml fresh culture medium then culture for another 24 h, the lactate concentration was measured (**g**), PPARγ and its fragment were analyzed by immunoblot (**h**). **i**, **j** THP-1 macrophages were cultured alone (Ctrl) or cocultured with MCF-7 (Coculture) in the presence or absence of PPARγ inhibitor GW9662. After 2 days, THP-1 macrophages were stained with Oil Red O and lipid content was quantified (**i**), or MCF-7 and regular culture medium was replaced with 2 ml fresh culture medium then culture for another 24 h, the lactate concentration was measured (**j**). All the histograms in this figure show means ± s.e.m. ($n = 3$, $*P < 0.05$, $**P < 0.01$, $***P < 0.001$, ns, not significant, one way ANOVA for multiple comparisons)

increased in THP-1 macrophages cocultured with MCF-7 cells compared with THP-1 macrophages cultured alone. In addition, the increase in lactate secretion occurred concomitantly with the onset of PPARγ cleavage at 24 h, suggesting that lactate secretion levels were directly associated with TAM differentiation (Fig. 4b). Consistently with these findings, the expression of the lactate transporter MCT4 expression was substantially increased in THP-1 cells cocultured with MCF-7 cells in a time-dependent manner

(Fig. 4c). This observation further confirmed that changes in lactate secretion were associated with TAM differentiation. Similarly, increased lactate secretion was observed in primary TAMs isolated from MMTV-PyMT mice, and this increase was significantly attenuated in primary TAMs derived from mice treated with YVAD (Fig. 4d). The expression of MCT4 exhibited a trend similar to lactate secretion (Fig. 4e), thus suggesting that increased lactate

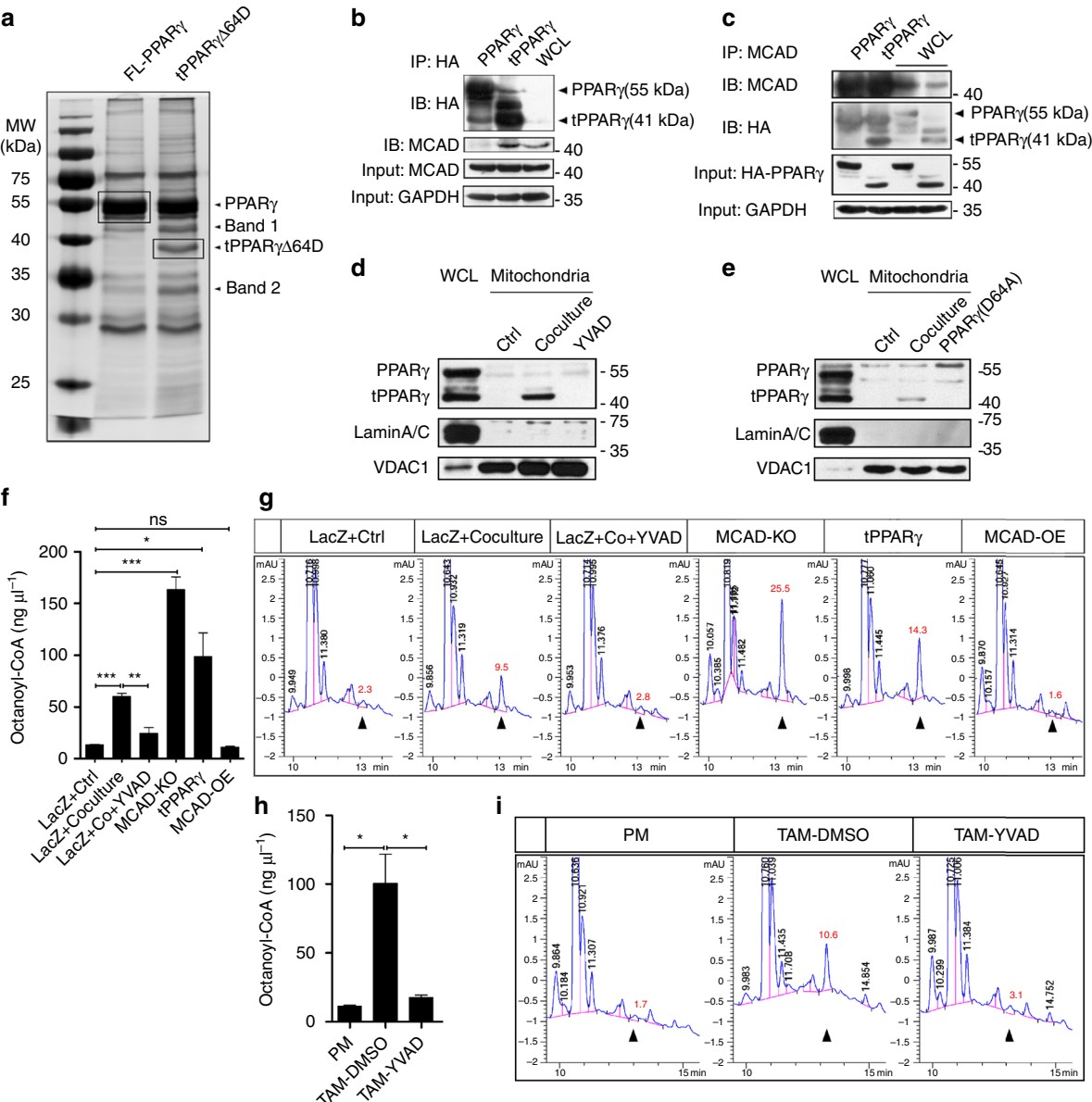

**Fig. 6** tPPARγ enriches in mitochondria and binds to MCAD and then impairs its activity. **a** A representative silver stained gel showing examples of the cellular proteins that specifically interact with tPPARγΔ64D (band 1 and 2). The corresponding Coomassie staining gel was shown in Supplementary Figure 28. **b**, **c** THP-1 cells were transfected with HA-PPARγ or HA-tPPARγΔ64D plasmids. Cell extracts were immunoprecipitated with anti-HA antibodies, then, analyzed with anti-HA antibodies or anti-MCAD antibodies (**b**), or the same cell extracts were immunoprecipitated with anti-MCAD antibodies, then, analyzed with anti-MCAD antibodies or anti-HA antibodies (**c**). MCAD expression or PPARγ expression in two transfected THP-1 cells or nontransfected cells was determined by immunoblot. **d**, **e** Extracts of the mitochondrial protein of THP-1 cells that cocultured with MCF-7 for 48 h in the absence or presence of YVAD (**d**), or transfected with mutant PPARγ (D64A) (**e**) were analyzed by immunoblot, laminA/C and VDAC1 as the loading control. PPARγ and tPPARγ in whole-cell lysates (WCL) in cocultured THP-1 macrophages was also shown. **f** LacZ overexpressing THP-1 cells were cultured alone or cocultured with MCF-7 in the absence or presence of YVAD for 24 h, MCAD knockout THP-1 macrophages, tPPARγ overexpression THP-1 macrophages (wild type PPARγ was deleted) and MCAD overexpression THP-1 macrophages were cultured alone for 24 h, then MCAD activity was analyzed by HPLC. Remaining octanoyl coenzyme A was negatively correlated with MCAD activity. Statistical results of total three trials were shown ($n = 3$). **g** One representative HPLC result was shown, the peak area of octanoyl coenzyme A was shown (mAU*s). **h** Peritoneal macrophages were isolated from tumor bearing MMTV-PyMT mice, and TAMs were isolated from tumor bearing MMTV-PyMT mice administration of vehicle or caspase-1 inhibitor YVAD, MCAD activity was analyzed by HPLC. TAMs in each sample were pooled from three mice ($n = 3$). **i** One representative HPLC result was shown. All the histograms in this figure show means ± s.e.m., ($n = 3$, *$P < 0.05$, **$P < 0.01$, ***$P < 0.001$, ns, not significant, one way ANOVA for multiple comparisons)

secretion might be associated with caspase-1 activation and TAM differentiation.

LDs are lipid-enriched cellular organelles[24]. The accumulation of LDs is typically associated with pathogenic statuses such as oxidative stress, tissue injury and mitochondrial dysfunction[25]. Oil Red O staining revealed that after 24 h of coculture, the level

of accumulated LDs in THP-1 cells increased significantly (Fig. 4f), indicating that the metabolic profile of THP-1 cells shifted during coculture with MCF-7 cells. Additionally, primary TAMs isolated MMTV-PyMT mice exhibited an 8-fold increase in LDs compared with peritoneal macrophages. Moreover, the administration of YVAD decreased LD levels in primary TAMs

(Fig. 4g). In addition, we observed heterogeneity in LD levels among cells within the same tumor tissue. After digesting tumor tissues into single cells, we found that a few of these cells exhibited excessive LD accumulation. The red arrow indicates cells with less LD accumulation, and these cells exhibited a morphology characteristic of tumor cells. The blue arrows indicate the cells that exhibited excessive LD accumulation (Fig. 4h). Notably, Oil Red O staining revealed that all of the isolated primary TAMs were overloaded with LDs (Fig. 4i). The morphological features of these TAMs closely resembled those of foam cells in atherosclerotic lesions, while macrophages isolated from the peritoneal of the same tumor bearing mice rarely detected with LDs (Fig. 4j). To quantitatively validate the cell-to-cell LDs heterogeneity within the tumor tissues, primary TAMs, pass-through cells and peritoneal macrophages were prepared, and analyzed by flow cytometry with lipophilic fluorescent dye (Nile Red). Consistently with the microscopy analysis, flow cytometry analysis confirmed that LDs accumulated in primary TAMs (Fig. 4k). Furthermore, high-resolution images of LD accumulation in primary TAMs, tumor cells and peritoneal macrophages isolated from MMTV-PyMT mice were evaluated. In contrast, we did not observe LD accumulation in single cells prepared from normal lung tissues (Supplementary Fig. 9c–e).

Most importantly, similar results were observed in cryosections of Her2[+] human breast tumor tissues. In these tissues, CD68[+] cells (TAMs) and Nile Red positive cells co-localized (Fig. 4l), thus further suggesting that excessive LD accumulation was presented in human breast tumor associated TAMs.

Because PPARγ cleavage strongly correlated with metabolic changes in TAMs and the cleavage of PPARγ impairs its transcriptional activity, we sought to determine whether PPARγ depletion would promote TAM differentiation and recapitulate the metabolic reprogramming observed in differentiating TAMs. Therefore, we inhibited PPARγ expression in THP-1 cells by using PPARγ shRNA. As expected, PPARγ knockdown failed to enhance TAM differentiation in our coculture model (Fig. 5a) and failed to upregulate either LD accumulation (Fig. 5b) or lactate secretion in THP-1 macrophages (Fig. 5c), possibly due to the fact that truncated fragment of PPARγ was attenuated too by inhibiting full-length PPARγ. The efficiency of PPARγ shRNA was verified by western blot analysis (Fig. 5d). Also we observed a similar effect in PPARγ knockout THP-1 cells and THP-1 cells that express PPARγ (D64A) mutant (wild type PPARγ was deleted). We tested pro-tumor genes expression and PPARγ activity (FABP4 expression) in coculture, we can see both PPARγ knockout and PPARγ (D64A) THP-1 cells downregulated pro-tumor genes expression, however, PPARγ activity is not impaired in PPARγ (D64A) THP-1 cells (Fig. 5e). We also examined the metabolic state (Fig. 5f, g), which maintains intact in PPARγ mutant cells. PPARγ expression in THP-1 cells was verified by western blot analysis (Fig. 5h).

More interestingly, PPARγ inhibitor GW9662 cannot decrease LD content or lactate secretion in coculture (Fig. 5i, j). All these data showed that "cleavage" is the primary cause of TAM differentiation and metabolic reprogramming but not PPARγ activity. And the traditional function of PPARγ is not involved in TAM differentiation. Therefore, it appears that the disruption of PPARγ activity does not share the same mechanism mediating TAM differentiation and metabolic reprogramming. It is of great interest to investigate whether metabolic reprogramming promotes TAM differentiation and, if so, whether and how PPARγ cleavage regulates metabolic reprogramming.

**Translocation of tPPARγ to mitochondria impairs MCAD activity.** To investigate the role of truncated PPARγ in TAM differentiation, we expressed HA-tagged full-length or

tPPARγΔ64D in HEK293T cells and performed an immuno-precipitation assay with an anti-HA antibody. The identification by mass spectrometry of 2 bands (35 and 45 kDa) specifically expressed in tPPARγΔ64D-transfected cells led to the identification of 4 potential tPPARγΔ64D binding partners with probability scores > 51 (MCAD, EF-Tu, SFXN and GNB2L1) (Fig. 6a and Supplementary Table 1). Among these proteins, medium-chain acyl-CoA dehydrogenase (MCAD) is involved in the mitochondrial fatty acid β-oxidation pathway, an important regulator of lipid metabolism. The potential interaction between tPPARγΔ64D and MCAD was further confirmed by coimmu-noprecipitation assays using HA-tagged full-length PPARγ- or tPPARγΔ64D-expressing THP-1 cells. Scanning analysis showed that the amounts of the MCAD protein input bound to tPPARγΔ64D increased eightfold compared with the full-length PPARγ (Fig. 6b, c), indicating that MCAD has a greater affinity for tPPARγΔ64D. Notably, EF-Tu and SFXN bind PPARγ and tPPARγΔ64D with a similar, lower affinity (Supplementary Fig. 10a, b). Importantly, 3 of the 4 putative tPPARγΔ64D-interacting proteins localized to mitochondria, suggesting that tPPARγΔ64D can translocate into mitochondria. We isolated mitochondria from 48 h cocultured THP-1 cells and found that tPPARγ was highly expressed in the mitochondrial fractions of cocultured cells. In addition, the administration of YVAD to cocultured cells prevented the trans-localization of tPPARγ to mitochondria (Fig. 6d). Also, tPPARγ was failed to present in mitochondria in cocultured THP-1 cells that express PPARγ (D64A; wild type PPARγ was deleted) (Fig. 6e). The enrichment of our isolated mitochondria was verified by immunoblot with markers for various organelles (Supplementary Fig. 10c). These experiments further confirmed that cleavage generated tPPARγ can translocate to mitochondria, which might indicate the unique functionality of tPPARγ in metabolic regulation.

MCAD is one of 4 similar enzymes (VLCAD, LCAD, MCAD, SCAD) associated with fatty acid β-oxidation, and MCAD deficiency leads to mitochondrial β-oxidation defects and metabolic disorders[26]. To explore whether the MCAD-tPPARγ interaction affects MCAD activity, we generated THP-1 cells stably overexpressing tPPARγΔ64D and MCAD and MCAD-deficient THP-1 cells by using lentiviral infection and the CRISPR/Cas9 system, respectively. The overexpression and knockout efficiency of these techniques was verified by western blotting (Supplementary Fig. 10d, e). We measured MCAD activity with HPLC and octanoyl-CoA as the substrate, and the retention time (13 min) was confirmed by using purified octanoyl-CoA standard (Supplementary Fig. 10f). The results of the HPLC analysis confirmed that MCAD activity was impaired in THP-1 cells cocultured with MCF-7 cells and that the caspase-1 inhibitor YVAD restored MCAD activity. As expected, the overexpression tPPARγΔ64D also reduced MCAD activity. MCAD overexpressing cells and MCAD knockdown cells were used as positive and negative controls (Fig. 6f), respectively and images of representative HPLC chromatograms are shown (Fig. 6g). However, coculture failed to inhibit MCAD activity in PPARγ deficient or PPARγ (D64A; wild type PPARγ was deleted) expressing THP-1 cells (Supplementary Fig. 10g). In addition, we confirmed that MCAD protein levels were not reduced in THP-1 cells cocultured with MCF-7 cells (Supplementary Fig. 10h). These findings were consistent with the observation that MCAD activity was severely impaired in TAMs derived from MMTV-PyMT mice, and this effect was rescued by the administration of YVAD every 2 days for three weeks in MMTV-PyMT mice (Fig. 6h, i). MCAD protein levels were unaffected in peritoneal macrophages and TAMs in vivo (Supplementary Fig. 10i). We also confirmed that MCAD expression does not interfere with PPARγ cleavage in coculture (Supplementary Fig. 10j). Together,

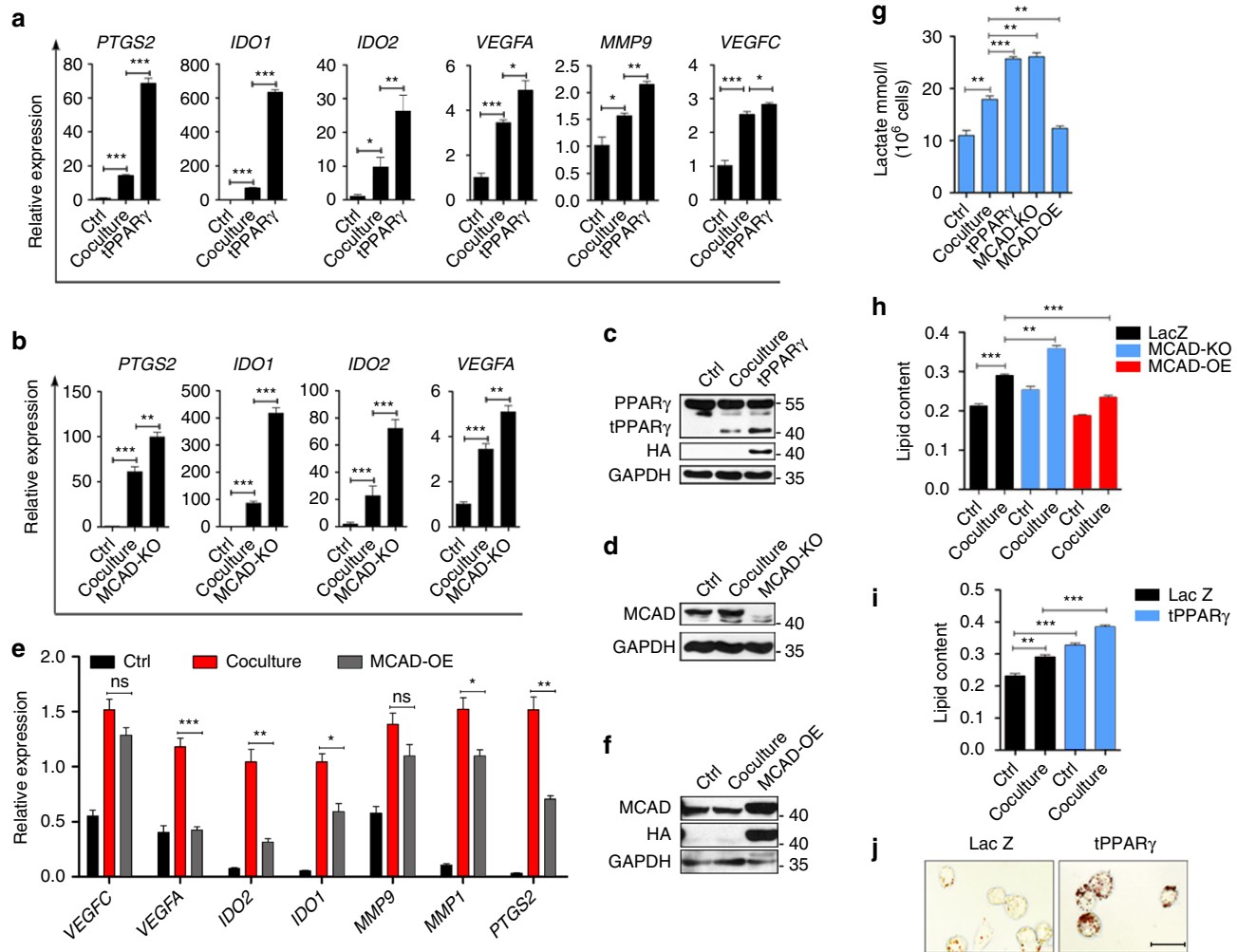

**Fig. 7** tPPARγΔ64D overexpression or MCAD knockout strongly potentiates TAM differentiation and recapitulates the metabolic reprogramming. **a** THP-1 macrophages cultured alone, THP-1 macrophages overexpressing LacZ or tPPARγ were cocultured with MCF-7 for 2 d, then TAM hallmarks were analyzed by RT-qPCR. **b** THP-1 macrophages cultured alone, LacZ overexpressing or MCAD knock out THP-1 macrophages were cocultured with MCF-7 for 2 d, TAM hallmarks were analyzed by RT-qPCR. **c, d** Immunoblot analysis of the tPPARγ overexpression (**c**) or MCAD knockout (**d**) efficiency. **e** THP-1 macrophages cultured alone (Ctrl), THP-1 macrophages overexpress LacZ (Coculture) or MCAD (MCAD-OE) were cocultured with MCF-7 for 2 d, TAM hallmarks were analyzed by RT-qPCR. **f** Immunoblot analysis of the MCAD overexpression efficiency. **g** THP-1 macrophages expressing LacZ (LacZ) were cultured alone (Ctrl) or cocultured with MCF-7 (Coculture) for 1 d, tPPARγ expressing (tPPARγ), MCAD knock out (MCAD-KO), MCAD overexpression (MCAD-OE) THP-1 macrophages were cocultured with MCF-7 (Coculture) for 1 day. Lactate secretion ability was measured. **h, i** LacZ (LacZ) overexpressing, MCAD knock out (MCAD-KO), MCAD overexpression (MCAD-OE) THP-1 macrophages (**h**) tPPARγ overexpression THP-1 macrophages (tPPARγ) (**i**) were cultured alone (Ctrl) or cocultured with MCF-7 (Coculture) for 1 day, LD content in macrophages were detected. **j** THP-1 macrophages overexpressing LacZ were cultured alone, tPPARγ overexpressing THP-1 macrophages were cocultured with MCF-7 for 1 d, then THP-1 macrophages were stained with Oil Red O, representative photographs were taken. Scale bars, 25 μm. Data in **a, b** were generated and analyzed by Step one plus Real-Time PCR-system (AB Applied biosystem). All the histograms in this figure show means ± s.e.m. ($n = 3$, *$P < 0.05$, **$P < 0.01$, ***$P < 0.001$, ns, not significant, one way ANOVA for multiple comparisons)

these findings indicate that MCAD is a key binding partner of the truncated PPARγ fragment of generated by caspase-1-mediated cleavage. This interaction appears to inactivate MCAD and might have significant effects on mitochondrial metabolism.

**tPPARγΔ64D promotes TAM differentiation and metabolic reprogramming.** Next, we explored the role of the MCAD-tPPARγ interaction in TAM differentiation. To determine if MCAD is required for tPPARγ-induced TAM differentiation, we examined the combined effect of tPPARγΔ64D overexpression and MCAD knockout on TAM differentiation. HA-tPPARγΔ64D overexpression in wild type or PPARγ knockout THP-1 cells strongly enhanced the expression of tumorigenic genes

compared with THP-1 cells overexpressing the control LacZ construct in the coculture model (Fig. 7a and Supplementary Fig. 11a). Similarly, MCAD knockout (MCAD-KO) in THP-1 cells markedly increased coculture-induced tumorigenic gene expression (Fig. 7b). Furthermore, HA-tPPARγΔ64D overexpression and MCAD knockout efficiency were evaluated by western blotting (Fig. 7c, d and Supplementary Fig. 11b). As expected, MCAD overexpression (MCAD-OE) suppressed the expression of tumorigenic genes (Fig. 7e). MCAD over-expression efficiency was verified by western blot analysis (Fig. 7f). Furthermore, MCAD-OE in coculture totally abolished the coculture-induced increase in lactate release and LD accu-mulation, and MCAD-KO further enhanced lactate secretion and LD accumulation (Fig. 7g, h). Moreover, the overexpression of

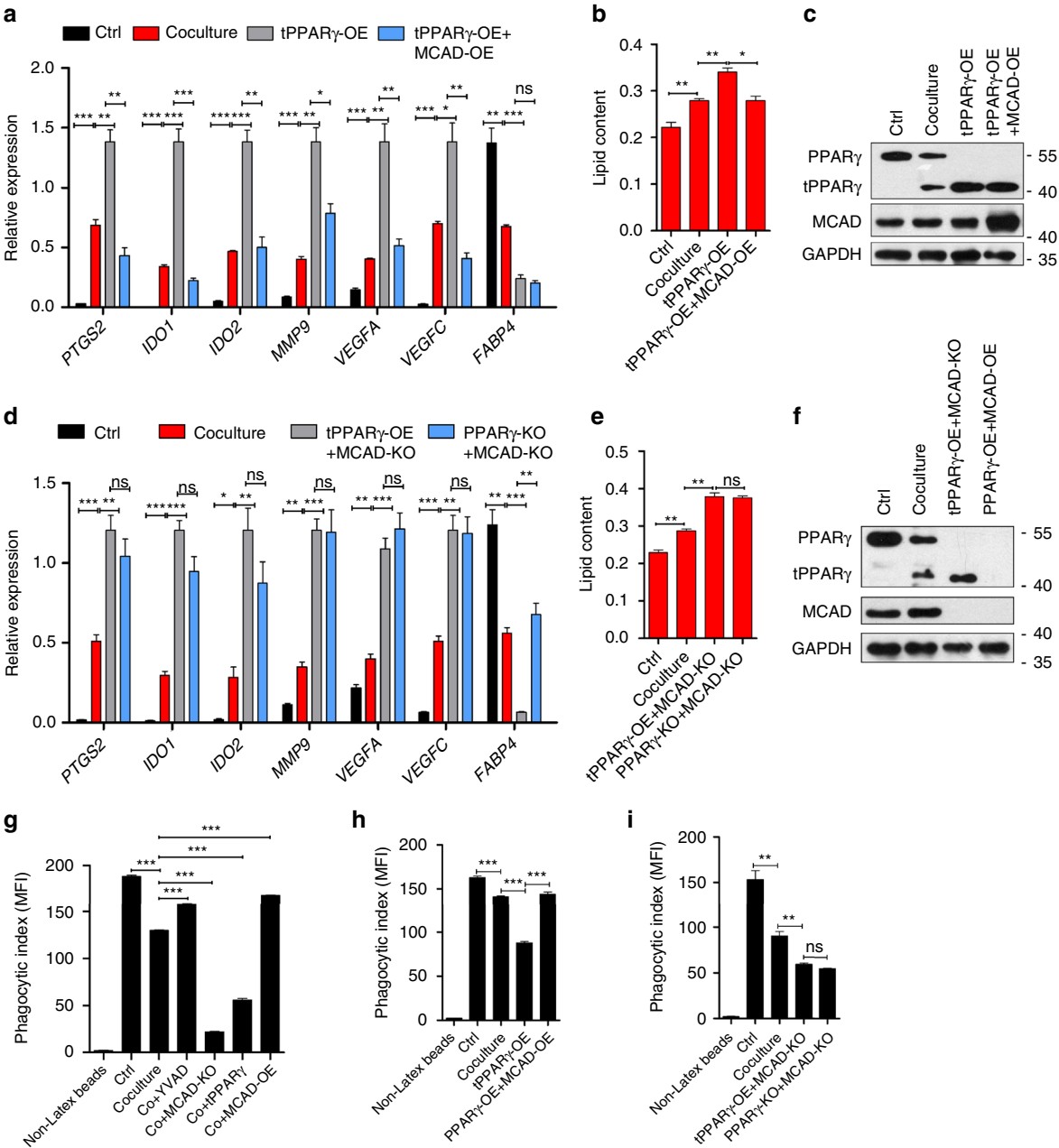

**Fig. 8** tPPARγΔ64D regulates TAM differentiation and metabolic reprogramming through MCAD. **a–c** LacZ overexpressing THP-1 macrophages were cultured alone or cocultured with MCF-7 for 2 d, THP-1 macrophages overexpressing tPPARγΔ64D or simultaneously overexpressing tPPARγΔ64D and MCAD were cocultured with MCF-7 for 2 d, TAM hallmarks were analyzed by RT-qPCR (**a**) and lipid content was analyzed by Oil Red O staining (**b**) PPARγ and MCAD expression were analyzed by immunoblot (**c**). **d–f** LacZ overexpressing THP-1 macrophages were cultured alone or cocultured with MCF-7 for 2 days, MCAD$^{-/-}$ THP-1 macrophages overexpressing tPPARγΔ64D or LacZ were cocultured with MCF-7 for 2 d, TAM hallmarks were analyzed by RT-qPCR (**d**) and lipid content was analyzed by Oil Red O staining (**e**), PPARγ and MCAD expression were analyzed by immunoblot (**f**). **g** LacZ overexpressing THP-1 cells were cultured alone or cocultured with MCF-7 in the absence or presence of YVAD for 2 d, MCAD knockout THP-1 macrophages, tPPARγ overexpression THP-1 macrophages (wild type PPARγ was deleted) and MCAD overexpression THP-1 macrophages were cocultured with MCF-7 for 2 d. Flow cytometry analysis of macrophage phagocytic index (phagocytosis of latex beads). MFI represents mean fluorescence intensity. **h** Flow cytometry analysis of macrophage (described in **a–c**) phagocytic index. **i** Flow cytometry analysis of macrophage (described in **d–f**) phagocytic index. All the histograms in this figure show means ± s.e.m. ($n = 3$, *$P < 0.05$, **$P < 0.01$, ***$P < 0.001$, ns, not significant, one way ANOVA for multiple comparisons)

tPPARγΔ64D further enhanced lactate secretion and LD accumulation (Fig. 7g, i, j).

To further determine whether tPPARγΔ64D effects on TAM differentiation is due only to its inhibition of MCAD, we performed a co-transfection in coculture with tPPARγΔ64D and MCAD, we found that MCAD reversed the pro-differentiation effect of tPPARγΔ64D (Fig. 8a) and differentiation-associated lipid metabolism (Fig. 8b). In addition, we found that TAM differentiation (Fig. 8d) and lipid content (Fig. 8e) were no longer sensitive to tPPARγΔ64D in MCAD$^{-/-}$ PPARγ$^{-/-}$ cells. Protein levels of PPARγ and MCAD were verified by western blot (Fig. 8c, f). These data convincingly showed that tPPARγΔ64D

regulates TAM differentiation through MCAD. PPARγ's target gene *FABP4* was used as a non-tumorigenic gene control.

To further explore whether tPPARγ affects mitochondrial function, we measured the oxygen consumption rates (OCRs) of THP-1 macrophages that overexpressing LacZ control or tPPARγ. Basal OCRs were measured, followed by serial additions of oligomycin (an inhibitor of ATP synthesis), carbonyl cyanide-ptrifluoromethoxyphenylhydrazone (FCCP; an uncoupling ionophore), and rotenone with antimycin A (blocking agents for complexes I and III of the electron transport chain, respectively) to discern the relative contributions of mitochondrial and non-mitochondrial mechanism of oxygen

consumption[27]. We found that tPPARγ mediated β-oxidation inhibition reduced both basal OCRs and maximal respiratory capacity in mitochondria (Supplementary Fig. 11c). Moreover, mitochondrial content in each group was analyzed by mitotracker fluorescent probes, the result demonstrated that OCRs changes were not due to mitochondria number changes but mitochondria functions (Supplementary Fig. 11d).

The phagocytic ability of TAMs is important for tumor cell phagocytosis, elimination and antigen presentation[28]. Here, we found that TAM's phagocytic ability was inversely associated with the levels of TAM differentiation, caspase-1 activity or expression of tPPARγ, and such ability was positively correlated with MCAD

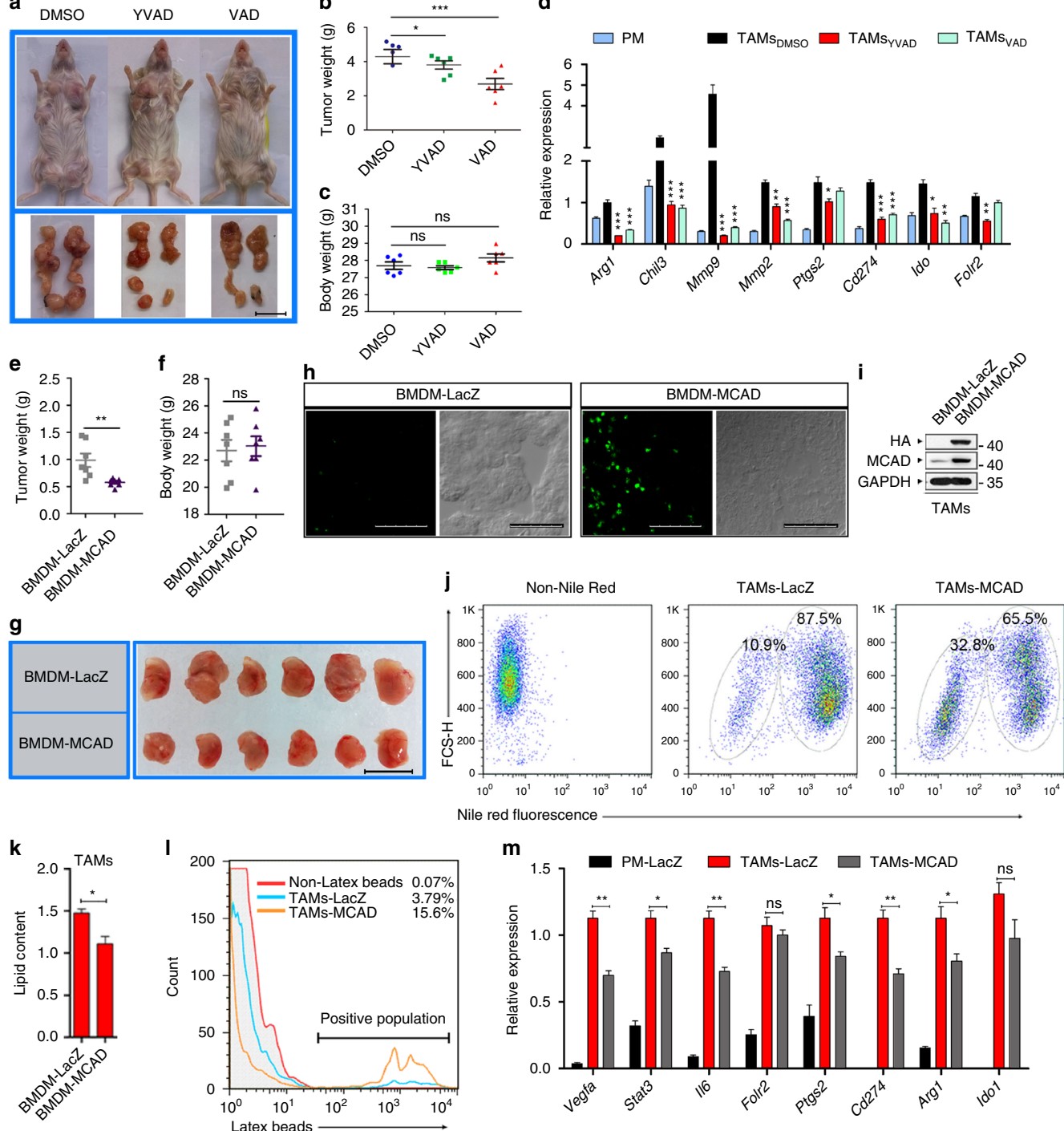

activity (Fig. 8g). Furthermore, in MCAD knockout TAM cells, over-expression or knockdown of tPPARγ lacks the regulation on TAM's phagocytic ability, suggesting that indeed tPPARγΔ64D regulates TAM's phagocytosis though MCAD (Fig. 8h, i).

These results suggested that PPARγ cleavage induced metabolic changes and promoted TAM differentiation by modulating MCAD activity in our coculture model, confirming that the caspase-1/PPARγ pathway plays a critical role in TAM differentiation.

**Caspase-1 inhibitors and mMCAD-transduced BMDMs suppress tumor growth.** Considering the inhibitory effect caspase-1 inhibitors exerted on tumorigenic TAM functions, we studied the anti-tumorigenic effect of YVAD and VAD in transgenic female MMTV-PyMT mice (approximately 8 weeks old). When tumor sizes grew to approximately 3–4 mm in diameter, DMSO, YVAD (5 mg kg$^{-1}$) or VAD (2 mg kg$^{-1}$) was intraperitoneally administered every other day for 3 weeks. Tumor growth was significantly inhibited in MMTV-PyMT mice treated with YVAD or VAD compared with DMSO-treated control (Fig. 9a, b). However, the body weight of mice and TAM content in tumor tissues in all of the experimental groups was unaffected (Fig. 9c and Supplementary Fig. 12a). Moreover, the TAM markers *Arg1*, *Chil3*, *Mmps*, *Ptgs2*, *Cd274*, *Ido1* and *Folr2* were significantly downregulated in TAMs derived from YVAD- or VAD-treated mice (Fig. 9d). Therefore, we conclude caspase-1 inhibitors are promising candidates for the clinical development of anti-breast cancer drugs.

We demonstrated that MCAD modulated LD accumulation, lactate secretion, and TAM differentiation. To validate the ability of MCAD to modulate TAMs in vivo, we engineered BMDMs overexpressing HA-tagged MCAD or a LacZ control. On day 4 after the tumor cells were injected (4T1), we intraperitoneally injected tumor-bearing mice with $1 \times 10^7$ MCAD-transduced BMDMs, or $1 \times 10^7$ LacZ-transduced BMDMs. The injections were repeated on day 13, and on day 27, the mice were sacrificed and the tumors were resected. We found that 2 injections of MCAD-transduced BMDMs significantly decreased 4T1 tumor growth, compared with the control mice ($P < 0.01$; Fig. 9e, g), and body weight was unaffected in both groups (Fig. 9f). BMDM infiltration into tumor tissues was confirmed by immunofluorescence assays (Fig. 9h) or western blot analysis with anti-HA or anti-MCAD antibodies (Fig. 9i). Nile Red staining confirmed that MCAD overexpression in BMDMs significantly reduced LD accumulation in TAMs derived from 4T1 mice, with an ~ 20% increase in lipid-free TAMs after infusion of MCAD-transduced BMDMs (Fig. 9j and Supplementary Fig. 12b). Moreover, the

quantification result of LDs content with Oil Red O was also shown (Fig. 9k). Furthermore, we found that phagocytic activity in TAMs derived from MCAD-OE BMDMs-treated mice was significantly upregulated compared with controls, the phagocytosis of latex beads (Fig. 9l and Supplementary Fig. 12c) and tumor cells (Supplementary Fig. 12d, e) by TAMs was demonstrated. Additionally, the expression of the TAM marker genes significantly decreased in MCAD-overexpressing TAMs (Fig. 9m).

In summary, MCAD over-expressing BMDMs significantly inhibited tumor growth in the 4T1 breast cancer mouse model, indicating that the anti-tumorigenic role of MCAD might be valuable in the development of novel anti-cancer immunotherapies.

## Discussion

In the current study, we provided novel insights into the mechanisms underlying TAM regulation and demonstrated that caspase-1/tPPARγ/MCAD signaling axis plays an essential role during TAM differentiation in vitro and in vivo (Fig. 10). We demonstrated that targeting macrophages via caspase-1 specific inhibitors or overexpressing MCAD in macrophages significantly attenuated tumor growth. Furthermore, we observed a strong correlation between lipid accumulation and TAM's tumorigenic function. It is noteworthy that a lipid content discrepancy between in vitro and in vivo system was observed (Fig. 4f, g), we speculate the accumulation of lipids in TAMs in vivo might be due to increased synthesis of fatty acids in TAMs or result from increased lipid uptake from extracellular or other origins. However, in in vitro system, the sources of lipid are far fewer than in vivo system, accumulation might be simply due to the synthesis mechanism within TAM cells. However, the precise cause of LD accumulation and the mechanism by which they promote TAM differentiation remain unknown. Therefore, additional studies are necessary to identify and characterize the specific lipid species that regulate distinct aspects of TAM differentiation associated with MCAD both in vitro and in vivo.

Metabolic reprogramming in macrophages by certain stimulators is an important characteristic feature of TAMs that primarily affects their extrinsic regulatory functions in the tumor microenvironment[29]. For example, several independent studies have recently highlighted the importance of metabolic coupling between cancer cells and other tumor stromal cells in promoting "parasitic cancer metabolism", in which tumor cells function as metabolic "parasites" that extract energy sources, including fatty acids, L-lactate, ketones, and glutamine, from supporting host cells. Such metabolic coupling promotes tumor cell growth

**Fig. 9** Caspase-1 inhibitors YVAD or VAD and mMCAD-transduced BMDMs suppress tumor growth. **a** Representative photographs of tumors in MMTV-PyMT mice. Scale bar, 2 cm. **b** Tumor weight was analyzed ($n = 6$). **c** Body weight of tumor-bearing mice was analyzed ($n = 6$). **d** Expression of TAM hallmarks was assessed by RT-qPCR ($n = 3$). **e, f** Tumor weight and body weight were measured ($n = 7$). **g** Representative photographs of tumors were taken. Scale bar, 2 cm. **h** Cryosections were prepared, the infiltration of MCAD-transduced BMDMs was observed via immunofluorescence with anti-HA antibodies (*green*). Scale bars, 75 μm. **i** Lysates of TAMs isolated from 4T1 tumor-bearing mice that infused with MCAD-transduced BMDMs mice or LacZ-transduced BMDMs were prepared, MCAD protein content was analyzed with an anti-HA antibody or anti-MCAD antibody. **j** Representative dot plots (logarithmic scale) of Nile Red fluorescence versus forward scatter (FCS-H) in TAMs isolated from 4T1 tumor-bearing mice that infused with LacZ-transduced BMDMs (TAMs-LacZ) or MCAD-transduced BMDMs (TAMs-MCAD) were shown, non-Nile Red stained TAMs isolated from 4T1 tumor-bearing mice that infused of LacZ-transduced BMDMs as control (Non-Nile Red). **k** Lipid content in TAMs isolated from 4T1 tumor-bearing mice that infused with MCAD-transduced BMDMs or LacZ-transduced BMDMs were analyzed by a quantification method ($n = 7$). **l** Phagocytic abilities of TAMs isolated from 4T1 tumor-bearing mice that infused with MCAD-transduced BMDMs (TAMs-MCAD) or LacZ-transduced BMDMs (TAMs-LacZ) were analyzed with latex beads, TAMs-LacZ incubated with the vehicle as the negative control (Non-Latex beads). **m** Pro-tumoral genes expression of TAMs isolated from 4T1 tumor-bearing mice that infused with MCAD-transduced BMDMs (TAMs-MCAD) or LacZ-transduced BMDMs (TAMs-LacZ) were analyzed with RT-qPCR, peritoneal macrophages isolated from 4T1 tumor-bearing mice that infused of LacZ-transduced BMDMs as control (PM-LacZ; $n = 3$). All the histograms in this figure show means ± s.e.m. ($n = 3$, *$P < 0.05$, **$P < 0.01$, ***$P < 0.001$, ns, not significant, one way ANOVA for multiple comparisons)

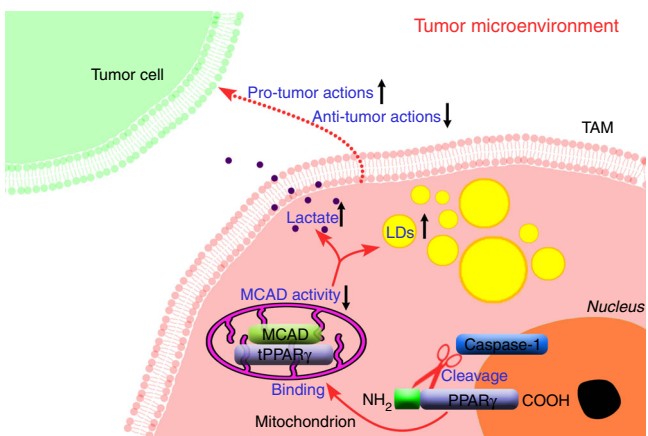

**Fig. 10** Schematic summary. In the current study, we provided novel insights into the mechanisms underlying TAM differentiation and demonstrated that caspase-1 activity increases during TAM differentiation, and that caspase-1 cleaves PPARγ at Asp64 and generates a 41 kDa fragment, this truncated PPARγ translocates to mitochondria and inactivates MCAD. MCAD inhibition causes lipid droplets (LDs) accumulation in TAMs and enhanced lactate secretion, MCAD activity and metabolic reprogramming are highly correlated with pro-tumor action of TAMs (*up arrow* for increase, *down arrow* for decrease)

and metastasis[29]. Cancer-associated fibroblast (CAF) secretes L-lactate, which fuels cancer cell growth and metastasis by initiating epithelial mesenchymal transition (EMT)[30]. The metabolic profile of TAMs has been shown to be associated with colorectal cancer, through a lipolytic co-activator, ABHD5[31]. Mechanistically, a recent study by Wenes et al. has shown that elevated glycolysis in TAMs contributes to angiogenesis in tumors. Such effects can be suppressed by mTOR inhibition, highlighting the role of macrophage metabolism in promoting tumor growth[32]. In agreement with previous studies, we found that genes that are differentially expressed during TAM differentiation are associated with metabolic pathways, thus suggesting that changes in metabolic reprogramming occur in differentiated TAMs. Therefore, it is important to consider both the intracellular and extracellular roles that metabolic reprogramming plays in the activation of oncogenic TAM functions.

It is well established that, caspase-1 primarily activates IL-1β and IL-18 in cancer. IL-1β and IL-18 are considered to be indicators of an increased risk of carcinoma and poor prognoses in multiple cancers[33]. However, activated caspase-1 targets a broad range of more than 100 cellular substrates, including ribosomal proteins, other caspases, heat shock proteins, cytoskeletal proteins, DNA replication licensing factors, ubiquitin-protein ligases, and so on[34]. The cleavage of these proteins elicits multiple changes in TAMs; however, the nature of these changes remains unknown. Our current investigation adds PPARγ to the list of caspase-1 substrates. This specific cleavage of PPARγ makes the cytosol-mitochondria translocation of truncated PPARγ, and initiates subsequent events, including the inhibition of MCAD and lipid accumulation, and eventually the reprogramming of TAMs. We have comprehensively linearized the caspase-1/tPPARγ/MCAD signaling pathway and provided the concrete evidence showing such signaling events play essential roles in reprogramming TAMs and leading to their differentiation.

Targeting macrophages has been evolved to be an effective therapeutic option for various diseases, including diabetes and cancer, as the presence and/or activity of macrophage are malleable in vivo in these disease models[35, 36] Developing such therapeutics targeting to minimize pathological processes will require the identification of unique pathways, as the direct abolition of macrophages is failed to suppress tumor growth[37–39]. Caspases inhibitor NCX-4016 has been extensively evaluated in treating colon cancer. Despite its ability to strongly induce apoptosis and inhibit cancer cell growth by targeting TAMs[40], NCX-4016 was still failed in clinical trials due to the unexpected fact that NCX-4016 can potentially induce genomic instability by releasing nitric oxide. In our current study, we showed that NCX-4016 impairs the pro-tumoral function of TAMs by suppressing the activity of caspases, especially caspase-1, just like YVAD and VAD which impair the tumorigenic functions of TAMs by suppressing caspase-1 activity, without exhibiting genotoxicity. Since activated caspase-1 form was only detectable in TAMs but not tumor cells or normal tissues (Supplementary Fig. 2d, e), caspase-1 inhibition will be relatively specific for TAMs. Reprogramming or repolarizing macrophages has been shown to be more effective than only blocking recruitment[41]. In our current research, both caspase-1 inhibitors do not display significant inhibitory effects on TAM's cell viability or growth, but leads to the disruption of PPARγ/MCAD activity and repolarization/reprogramming of TAMs, eventually suppressing the tumor growth. In conclusion, our findings shed light on the novel mechanism of TAMs in promoting tumor growth, and might facilitate the development of anti-cancer drugs with maximal anti-tumor properties and minimal side effects.

## Methods

**Reagents**. RPMI 1640 and the Dulbecco's modified Eagle's medium (DMEM) were purchased from Gibco. DCFH-DA probe was from Beyotime (Shanghai, China). Anti-mouse/human PPARγ (sc-7273, sc-7196, the dilution ratio was 1:500), anti-mouse caspase-1 (G-19, sc-1597, the dilution ratio was 1:1,000), MCAD (sc-98926 Lot # G2809, the dilution ratio was 1:1,000), EF-Tu (sc-393924, Lot # B0514, the dilution ratio was 1:1,000), SFXN1 (sc-160797, Lot # F0311, the dilution ratio was 1:1,000) and HA (sc-7392, Lot # E0614, the dilution ratio was 1:1,000) antibodies were purchased from Santa Cruz Biotechnology, Dallas, TX, USA. Anti-mouse FITC-F4/80 was purchased from eBioscience (Ref: 11-4801, the dilution ratio was 1:1,000). Anti-human caspase-1 antibody (D7F10, the dilution ratio was 1:1,000) was from Cell Signaling (Boston, USA). HRP conjugated goat anti-mouse/rabbit antibodies were from Beyotime (Haimen, China, the dilution ratio was 1:10,000). All cytokines such as M-CSF were obtained from Peprotech Systems (Minneapolis, MN). The pan-caspase inhibitor z-VAD-FMK, caspase-1 inhibitor z-WEHD-FMK and the caspase-3 inhibitor z-DEVD-FMK were obtained from R&D systems. The caspase-1 inhibitor Ac-YVAD-CMK was obtained from Bachem. NCX-4016 (NO-aspirin) were purchased from Nicox (Sophia Antipolis, France). All of other chemicals were purchased from Sigma-Aldrich unless otherwise described.

**Cell culture**. The THP-1(GDC100), HEK293T (GDC187) and MCF-7 (GDC055) human breast cancer cell lines and the 4T1 (GDC0294) murine mammary cell line were obtained from China Center for Type Culture Collection (Wuhan, China). The HUVEC cell line (ATCC, PCS-100-010) was obtained from Shanghai Bogoo Biotechnology. Co., Ltd (Shanghai, China) and cultured according to the manufacturer's instructions. All media were supplemented with 1% penicillin-streptomycin and 10% fetal bovine serum (Gibco, mycoplasma contamination check was carried out by our group).

**Mice**. Nine weeks old female BALB/c mice and nude mice were purchased from the Model Animal Research Center of the Nanjing University, Nanjing, China and bred in our animal facilities under specific pathogen-free conditions. Nine weeks old female *Caspase-1*−/− mice in the C57BL/6 background[42] (Originally from Dr. Vishva Dixit, Genentech) were generously provided by Prof. Feng Shao (National Institute of Biological Sciences, Beijing, 102206, China). All experiments related to animals were approved by Institutional Animal Care and Use Committee, Nanjing University.

**RT-qPCR**. Reverse transcription was performed with 5 × All-In-One RT MasterMix (abm Cat#G486 Code Q111-02) kit. And quantitative PCR reactions were performed using SYBR Green Master Mix (Vazyme biotech) kit, most data were collected and analyzed by CFX96 Real-Time System (BIO-RAD), a few by Step one plus Real-Time PCR-system (AB Applied biosystem) and specifically noted in figure legends. Primers used for qPCR are shown in Supplementary Table 2.

**Electrophoretic mobility shift assay (EMSA)**. Electrophoretic mobility shift assays (EMSA) were conducted with the LightShift Chemiluminescent EMSA Kit (Pierce) according to standard procedures. The sequences of the DNA probes to the wild-type and mutated binding sites as follows: PPRE, 5′-CAAAACTAGGT-CAAAGGTCA-3′ and the mutant is 5′-CAAAACTAGCACAAAGCACA (biotin-labeled).

**Caspase activity assay**. Caspase activity was assessed using caspase-1 (C1102), caspase-3 (C1115) and caspase-6 (C1135) activity fluorescence detection kit (Beyotime Institute of Biotechnology, Haimen, China) according to the manufacturer's instructions.

**Luciferase reporter assay**. HEK293T cells were transfected with plasmids using Lipofectamine 2000 (Invitrogen) according to the manufacturer's protocol. The plasmids used in the luciferase reporter assays included PPARγ-WT, PPARγ 64D/A mutant, tPPARγΔ64D mutant, the inflammasome-associated plasmids expressing NLRP3, ASC and caspase-1, and an internal control (pRL-SV40). The total quantity of the expression vectors used was equivalent to the quantity of the empty control vector used. At 24 h after transfection, luciferase and Renilla luciferase activities were measured using the Dual Luciferase Reporter assay system (Promega, Madison, WI) according to the manufacturer's protocol. The transfection efficiency was normalized to the Renilla luciferase activity. Experiments were conducted 3 times independently.

**Cytokine release assay**. Cell culture supernatants were analyzed with ELISA kits for the detection of IL-1β (Thermo Scientific, Cat#88-7261-22), IL-6 (Thermo Scientific, Cat#88-7066-22), IL-18 (Thermo Scientific, Cat#KHC0181) and VEGF (Thermo Scientific, Cat#KHG0112) according to the manufacturer's protocols.

**Coculture and preparation of concentrated conditioned medium (CCM)**. Approximately $1 \times 10^5$ THP-1 cells were seeded in six-well plates and treated with 50 nM PMA for 24 h to induce the cells to differentiate into THP-1 macrophages. The PMA-containing medium was removed after 24 h and the cells were washed three times in PBS to remove the PMA. PMA-treated THP-1 macrophages in six-well plates were cocultured with MCF-7 cells that had been left to attach to the inserts for 12 h before the coculture. The cells were cocultured in RPMI1640 supplemented with 10% FBS and 1% penicillin-streptomycin in the presence of the vehicle control, 50 μM YVAD, 50 μM VAD or 50 μM NCX-4016 for 48 h. Then, the MCF-7 cells and the regular culture medium were discarded and replaced with 2 ml of serum-free RPMI1640, and the cells were cultured for an additional 24 h. The conditioned medium was collected and concentrated 10-fold with ultrafiltration conical tubes with membranes selective for molecules < 10 kDa (Millipore UFC501096). The cells cultured with the vehicle control, 50 μM YVAD, 50 μM VAD and 50 μM NCX-4016 were labeled $CCM_{Coculture}$, $CCM_{YVAD}$, $CCM_{VAD}$ and $CCM_{NCX-4016}$, respectively, and the concentrated conditioned medium generated by THP-1 macrophages cultured alone was labeled $CCM_{Ctrl}$.

**Pro-angiogenesis assay**. Human umbilical vein endothelial cells (HUVECs) were used in the in vitro capillary morphogenesis assays. HUVECs were preincubated for 24 h with $CCM_{Ctrl}$, $CCM_{Coculture}$, $CCM_{YVAD}$, $CCM_{VAD}$ or $CCM_{NCX-4016}$ medium. The cells were subsequently seeded in 48-well plates coated with Matrigel ($3 \times 10^5$ cells per well). After 12 h at 37°C, the cells were fixed with glutaraldehyde and stained with Giemsa. The average number of tubules was calculated from images of three separate microscopic fields (× 40) and representative images were captured.

**Pro-invasion assay**. MCF-7 cells were seeded in Matrigel-coated invasion chambers (24 wells, 8 mm pore size) and cultured with $CCM_{Ctrl}$, $CCM_{Coculture}$, $CCM_{YVAD}$, $CCM_{VAD}$ or $CCM_{NCX-4016}$. After 12 h, MCF-7 cells were fixed and stained with crystal violet, and invading cells were subsequently counted.

**Pro-migration assay**. MCF-7 cells were cultured in 12-well plates ($3 \times 10^5$ per well) as confluent monolayers. The monolayers were serum-starved for 24 h and scratched in a line across the well with a 10 μl standard pipette tip. The wounded monolayers were then washed twice with serum-free media to remove cell debris and cocultured with $CCM_{Ctrl}$, $CCM_{Coculture}$, $CCM_{YVAD}$, $CCM_{VAD}$ or $CCM_{NCX-4016}$. Images of the same location of a cell-free wound were recorded under a microscope at 0 and 12 h.

**Preparation of murine peritoneal macrophages**. Peritoneal macrophages were isolated according to the methods used by Xia Zhang, Ricardo Goncalves and David M. Mosser[43].

**Isolation TAMs by MACS separation**. Tumor tissues were prepared as single cell suspensions, by minced in the DMEM with 0.1% collagenase I (1 h, 37 °C), passed through a 19 G needle, and filtered using a 40-μm cell strainer (BD Falcon). Cells in suspensions were stained for 30 min at 4 °C with anti-mouse F4/80 antibody. After

rigorous washing with 0.5% BSA and 2 mM EDTA in PBS, cells were incubated with an anti-FITC Microbeads (Miltenyi Biotec) for 20 min at 4 °C. F4/80+ cells were sorted by MACS separation according to the manufacturer's instruction (Miltenyi Biotec), then F4/80+ cells (TAMs) and pass-through cells (mainly tumor cells) were prepared.

**Lactate secretion measurement**. THP-1 Cells were cocultured with MCF-7 for indicated time then MCF-7 of regular culture medium was discarded and replaced with 2 ml fresh culture medium then culture for another 24 h, the lactate concentration was measured by lactate assay kit (Nanjing Jian Cheng, China, Ref # A019-2).

**Lipid droplets content measurement**. THP-1 macrophages ($5 \times 10^5$ cells per well) at different treatments were rinsed in PBS once (1 min) and then fixed in 4% paraformaldehyde for 10 min. After rinsing with PBS for 3 times, 3 min each, cells were stained with filtered Oil Red O (Sigma, Oo625-25G lot # SLBC9102V) working solution at room temperature for 2 h. Next, cells were rinsed again with 60% isopropanol. After a copious wash with distilled water, cells were then photographed under a microscope (Olympus) at ×200 magnification. For lipid droplets content quantification, Oil Red O dye was extracted with DMSO and optical density (OD) was detected using a spectrophotometer at 496 nm.

**Immunostaining**. Cryosections of Her2+ human breast tumor tissues were fixed with 4% paraformaldehyde for 30 min, washed in phosphate-buffered saline (PBS) and blocked with 5% BSA for 30 min. The slides were incubated with anti-CD68 (1:100, eBiosience ref: 11-0689) overnight at 4 °C, stained with DAPI to label nuclei for 15 min and incubated with Nile Red for 15 min at RT. After the cells were washed, they were evaluated with a fluorescence microscope.

All human tissues were obtained from Chinese PLA General Hospital, Beijing. All patients provided written informed consent in accordance with the declaration of Helsinki before enrolling in the study. The protocol in this study was approved by the institutional review board at the Chinese PLA General Hospital, Beijing. No commercial sponsor was involved in this study.

**Immunoprecipitation and western blot analysis**. For the immunoprecipitation assays, whole-cell extracts were prepared after transfection, incubated overnight with the appropriate antibodies and subsequently incubated with Protein A/G beads (Beyotime). The beads were washed 5 times with low-salt lysis buffer, and the immunoprecipitates were eluted in 1 × SDS Loading Buffer (Beyotime) and resolved by SDS-PAGE. Proteins were transferred to PVDF membranes (Bio-Rad) and incubated with the appropriate antibodies. The LumiGlo Chemiluminescent Substrate System (Thermo) was used for protein detection. All original western blotting blots are presented in Supplementary Figs 13–27.

**Mass spectrometry**. After immunoprecipitation, the protein bands were excised in our laboratory by using trypsin-based in-gel protein digestions, and the subsequent MS analysis was conducted at the School of Life Sciences, Peking University.

**Determination of MCAD activity by using HPLC**. To determine MCAD activity, macrophage pellets were re-suspended in PBS and homogenized by sonication. The protein concentration of the homogenates was determined using the bicinchoninic acid assay and human serum albumin was used as a standard. The standard mixture contained 250 mM Tris-HCl (pH 8.0) and 1.25 mM ferrocenium hexafluorophosphate (FcPF6) plus 450 μM octanoyl coenzyme A, in a final volume of 100 μl. The reaction was initiated by the addition of 80 μl of the standard mix to 20 μl of macrophage homogenate (1 mg ml$^{-1}$ final protein concentration) and was allowed to proceed at 37 °C. After 10 min, the reaction was stopped with 10 μl of 2 M HCl, and samples were neutralized with 10 μl of a solution containing 2 M KOH. After neutralization, 10 μl of 10 mM L-cysteine and 30 μl acetonitrile was added to the mixture. L-cysteine is added to reduce oxidized ferrocenium hexafluorophosphate, which may interfere with the chromatographic separation. The samples were centrifuged at 20,000 g for 5 min, and the remaining octanoyl coenzyme A in the supernatant was analyzed by HPLC with an elution system of acetonitrile and 16.9 mM sodium phosphate buffer pH 6.9. HPLC procedures were performed in State Key Laboratory of Pharmaceutical Biotechnology, Nanjing University.

**CRISPR-CAS9 knockout**. gRNA designed to target the common exons for all human MCAD isoforms were synthesized as follows: oligo #1, 5′-CACCGTGCTCGTAAATTTGCCCAGAG-3′, oligo #2, 5′-AAACCTCTGG-CAAATTTACGAGC AC-3′, and cloned to lentiCRISPRv2 (one vector system) plasmids[44]. gRNA designed to target the common exons for all human PPARγ isoforms were synthesized as follows: oligo #1, 5′-CACCGCTCCGTG-GATCTCTCCGTAA-3′, oligo #2, 5′-AAACTTACGGAGAGATCCACGGAGC-3′.

**Generation of lentiviral vector encoding the short hairpin RNA (shRNA)**. Oligonucleotide sequence 5′-GTGAAGAGATCCTTCTGTA-3′ that targeting the 3′-UTR of the human caspase-1, oligonucleotide sequence

5′-GTTTGAGTTTGCTGTGAAG-3′ against human PPARγ transcript (nucleotides 1095–1113), and a scrambled oligonucleotide 5′-GAGTGAGTAATTCATCCTG-3′ were inserted into pLL3.7 to generate shRNA plasmid. Packaging plasmids VSVG and Δ8.9 were used for lentivirus production and infection in 293T cells.

**Construction of lentiviral vectors.** We introduced the cDNA encoding the target genes of interest into the pLenti6/v5 lentiviral vector. The lentiviruses were produced in HEK293T cells by co-transfecting the target plasmid with 2 package plasmids (VSVg and Δ8.9) using Lipofectamine 2000 (Invitrogen). To enhance PPARγ (D64A) expression efficiency in PPARγ wild type knockout THP-1 cells, the synonymous mutation on PPARγ coding sequence (gRNA binding site) was generated: (nucleotides 58–72) 5′-GTGGATCTCTCCGTA-3′ mutated to 5′-GTTGATTTGTCCGTG-3′ by using Site-Directed Mutagenesis Kit (FMTG-25, SBS Genetech, Beijing, China).

**Phagocytosis.** Phagocytosis assays were conducted using fluorescent red latex beads (1 μM diameter, L-2778, Sigma-Aldrich). Latex beads were opsonized with complete medium (10% FBS in PRMI 1640) for 1 h at 37 °C before the phagocytosis assays. Opsonized beads were added to macrophage cells at a ratio of 10:1 and incubated at 37 °C for 4 h. Phagocytosis was terminated with the addition of 1 ml ice-cold sterile PBS. Cells were harvested and washed in ice-cold PBS 3 times and analyzed by using flow cytometry.

**Ex vivo phagocytosis assay.** TAMs were isolated by MACS with anti-CD11b Microbeads (BD Falcon) from tumor bearing MMTV-PyVT mice. $5 \times 10^5$ purified TAMs were plated per well in a 24-well culture plate. 4T1 tumor cells were labeled with 2.5 μM carboxyfluorescein succinimidyl ester (CFSE) according to the manufacturer's protocol (Invitrogen). Macrophages were incubated in serum-free medium for 2 h before adding $2 \times 10^5$ CFSE-labeled live tumor cells and incubated for 4 h. Macrophages were repeatedly washed and phagocytosis was then determined by flow cytometry detection of CD11b$^+$ (PE labeled) TAMs. The phagocytic index was calculated by mean fluorescence intensity (MFI) of CFSE.

**Flow cytometry analysis.** The cells were analyzed on a fluorescence-activated cell sorting (FACS) Calibur cytometer using Cellquest software (Becton Dickinson) and FlowJo software (V. 7.6.4, TreeStar). The statistics presented are based on at least 10,000 events gated on the population of interest.

**Tumor models.** When confluence reached 80%, cultured 4T1 cells were harvested from the monolayer cultures, washed with serum-free medium, and re-suspended in RPMI 1640, $1 \times 10^7$ cells per ml, and then 20 μl of cell suspension was injected into the inguinal mammary fat pads of syngeneic, immunocompetent BALB/c mice.

**YVAD and VAD administration.** YVAD or VAD was intraperitoneally injected into tumor-bearing mice. The dosage used in this study was 5 mg kg$^{-1}$ for YVAD and 2 mg kg$^{-1}$ for VAD.

**Generation of retrovirally transduced macrophages from bone marrow cells.** BMDMs were generated according to the methods used by Xia Zhang, Ricardo Goncalves and David M. Mosser[43]. Fresh medium containing concentrated lentivirus that directs expression of MCAD gene (by using PEG-it Virus Precipitation Solution, CAT # LV810A-1) was added to the plates, after 3 days transduced macrophages were enriched by blasticidin for another 3 days and then cells were harvested, and then washed in RPMI 1640 before injection.

**Nude mice tumor model.** A total of $1 \times 10^6$ C57BL/6-derived breast cancer cell line EO771 cancer cells in Matrigel (Corning, Cat No. 354230) were implanted in the flank alone or in combination with $2.5 \times 10^5$ WT or caspase-1-deficient (KO) BMDMs. Tumors were resected and measured 21 days later.

**Lung cell isolation.** After perfusion with PBS containing heparin (1 g l$^{-1}$), the right lung of mice was harvested, diced, and incubated on an orbital shaker for 90 min at 37 °C in digestion medium (DMEM medium containing 10% fetal bovine serum, 300 U ml$^{-1}$ collagenase type, 50 U ml$^{-1}$ DNase, 100 U ml$^{-1}$ penicillin and 100 μg ml$^{-1}$ streptomycin). After vortexing for 10–15 s, the product of digestion was passed through 40 μm cell strainer. Cells were collected after 5 min centrifugation at $650 \times g$. The cell pellet was washed with PBS for three times then re-suspended in DMEM medium containing 10% fetal bovine serum. Cells were cultured in the dish for 12 h, then, both attaching and floating cells were stained with Oil Red O, and analyzed by microscopy.

**Microarray analysis.** The microarray platform used in this study was the Affymetrix GeneChip PrimeView Human Gene Expression Array. The microarray analysis was conducted by Gminix (Shanghai, China).

**Pathway analysis.** Pathway analysis was used to find out the significant pathway of the differential genes according to KEGG, Biocarta and Reatome. Still, we turn to the Fisher's exact test and $\chi^2$ test to select the significant pathway, and the threshold of significance was defined by P-value and FDR (Kanehisa M et al. 2004; Yi M et al. 2006; Draghici S et al. 2007). The enrichment Re was calculated like the equation Re = (nf/n)/(Nf/N) (Schlitt T et al. 2003).

**Experimental repeats and statistics.** All immunoblotting assays had been performed at least thrice to ensure the repeatability of the experiments, and the representative pictures were shown in the final figures. TAMs cells were isolated and pooled from at least two mice for every band in western blot or every sample in flowcytometry or RT-qPCR. All statistical analysis was generated based on at least three independent experiments. Data for all experiments were analyzed with Prism software (GraphPad).

**Data availability.** The microarray data have been deposited in GEO under the accession codes GSE99960. Other data that support the findings of this study are available from the article and its Supplementary Information files.

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

## Acknowledgements

We thank Prof. Feng Shao (National Institute of Biological Sciences, Beijing, 102206, China) for kindly providing *caspase-1*⁻/⁻ mice. We thank Prof. Gang Hu, Dr. Min Lu at Nanjing Medical University for the nice gift of several KO mouse strains. Thanks to Prof. Songshan Jiang at Sun Yat-sen University for providing us certain plasmids. We thank members of Shen's laboratory for helpful discussions, technical assistance and critical reading of the manuscript. This manuscript has been revised and polished by Nature Publishing Group. The work was supported by the National Natural Science Foundation of China under Grant 81273527, 81421091, the National Key Research and Development Program of China (2017YFA0205400) and partially supported by the National Natural Science Foundation of China (No. 81473220, 81673439, 91013015). Q.S. is supported by JDRF advanced postdoc fellowship award (3-APF-2016-205-A-N), USA.

## Author contributions

Z.N., Q.S. and P.S. designed this study and wrote the paper. Z.N., Q.S., W.Z., Y.S, and N.Y. performed the experiments. J.J., Q.W. and X.Z. conducted the MS experiments and contributed to the discussion. The rest of the work mentioned in this study was contributed by other authors. P.S. supervised the project. All authors critically reviewed the article and approved the final manuscript.

## Additional information

**Competing interests:** The authors declare no competing financial interests.

