## [Peer Review File · Nature Communications]

Reviewers' comments:

Reviewer #1 (Remarks to the Author):

Nie et al demonstrate that caspase 1 is a central regulator in tumor associated macrophage differentiation to a tumor promoting phenotype. The authors show that caspase 1 promotes TAM differentiation by cleaving PPAR γ to a truncated form, which localizes to the mitochondria, where it then binds with and attenuates MCAD. The interaction between truncated PPAR γ and MCAD inhibits fatty acid oxidation leading to accumulation of lipid droplets and promotes TAM differentiation. The authors provide evidence of therapeutic utility of targeting TAMs in cancer therapy because they demonstrate caspase 1 inhibitors or IP injections of BMDM overexpressing MCAD suppresses tumor growth in two different mouse models of breast cancer. Modulating TAMs for cancer therapy is of high clinical interest and the group provides sufficient evidence that caspase 1 and/or MCAD could be targets for clinically modulating TAMs.

The extent of the in vitro analysis are impressive and the in vivo TAM validations and confirmation strengthens the story. Overall the authors are commended for a very extensive and thorough in vitro analysis. There are several minor points that need to be addressed prior to publication as detailed below. It was difficult to tell how many times each experiment was performed. The methods claim that statistics were calculated based on at least 3 independent experiments - but for experiments without statistics (western blots, RNA, for example) it is not clear how many times the experiment was performed. For the mouse experiments it appears that 5-7 mice were enrolled per treatment group and it appears that each experiment was only done one time; however given the experiments were done in two different mouse models, this may be sufficient.

1. Figure 1 shows that caspase 1 inhibition can block PPAR γ cleavage in TAMs, indicating that caspase 1 mediates PPAR γ cleavage in TAMs. Confirming the co-culture results with TAMs isolated from caspase 1 treated mice is nice.

a. Figure 1c - Since the many following studies rely on the isolation of TAMs from the tumors please provide confirmation by flow or IF that the "sorted TAMs" are of high purity macrophages (>95% F480). Also, for Figure 1c, where did these cells come from 4T1 or MMTV?

b. There is no mention for how many times each western blot was run and how many mice were used in each experiment. Please provide details on sufficient number of repeats for western blots and TAM isolations in figure 1 (and throughout manuscript).

2. In supplementary Figure 2e the authors state "overexpression of HA-NLRP3 significantly increased the levels of the truncated form of PPAR γ and concomitantly decreased the expression of pro-caspase 1". Please provide (1) statistics and (2) a western blot with better separation, to see the tPPAR γ .

3. Figure 2 - the authors show that caspase 1 inhibition reverses PPAR γ cleavage and blocks TAM differentiation, as measured by RT-qPCR and cell based assays. While there is a statistically significant reduction in the "TAM phenotype" with caspase inhibitors, the phenotype is not entirely rescued. What accounts for the partial rescue?

4. In supplementary figure 4 - how many times were each of these experiments performed? Taking 3 pictures of one well is not sufficient to generate statistics (S4a).

5. Supplemental figure 5: IL-1 β and IL-18 are found in the cell culture media of the co-culture system. The authors wanted to test if either of the caspase 1 substrates played a role in TAM differentiation. Adding exogenous IL-1 β or IL-18 induces THP1 differentiation similar to MCF7 co-culture, but the authors conclude these cytokines are not involved in TAM differentiation. This is a confusing figure and conclusion.

6. Figure 7 provides clinically relevant findings in that caspase inhibitors can cause tumor shrinkage in the MMTV-PyMT breast tumor animal model. They show this response is macrophage dependent because IP injection of MCAD over expression BMDM similarly causes tumor shrinkage in the 4T1 tumor model.

a. Please quantitate and provide statistics for 7J and 7L.

b. It's interesting that the TAMs-MCAD are more phagocytic (7L) are there other markers that can show these macrophages are more "anti-tumor" like? Are they engulfing tumor cells? Do they

increase co-stimulatory molecules (CD40 is shown in the microarray, was this confirmed at the protein level?). Again a decrease in "TAM differentiation" markers are shown in 7m, but is there an increase in anti-tumor markers?

c. The authors should show that IN VIVO treatment of caspase 1 inhibitors decreases PPAR γ cleavage and alters lipid accumulation in TAMs. I cannot see that this is ever done.

7. A schematic would be helpful to the reader to put the whole story together.

8. What strain of MMTV-PyMT mice are used?

9. This is beyond the scope of the work, but could the author speculate: What is the secreted MCF7-derived factor that induces the macrophage differentiation to a pro-tumor phenotype?

Reviewer #2 (Remarks to the Author):

In the present manuscript, Niu et al studied the promoting roles of caspase-1 in Tumor-associated macrophages (TAMs) differentiation by cleaving peroxisome proliferator-activated receptor gamma (PPAR γ) at Asp64. The truncated 41 kDa PPAR γ translocates directly interacts with medium-chain acyl-CoA dehydrogenase (MCAD) in mitochondria and inhibits fatty acid oxidation, thereby promoting TAMs differentiation. They propose that targeting the caspase-1-PPAR γ - MCAD pathway might be a promising therapeutic approach to prevent tumor progression. It is interesting. However, the following issues should be addressed.

1. The untreated THP-1 macrophages or the untreated macrophages isolated or derived from other sources were used control in the whole manuscript. The standardized M1 (LPS+IFN-g) and M2 (IL-4) macrophages should be used as controls as well, so that the authors can conclude the unique roles of caspase-PPAR γ in TAM differentiations. These are essential controls.

2. Specific inhibitors of caspase-3 and caspase-6 attenuate the cleavage of PPAR γ protein in response to TNF α in cultured adipocytes (ref 15). The positive controls for inhibitors on caspase-3 and caspase-6 are needed in Fig. 1h and Fig. 1i.

3. The soluble factors released by tumor cells to trigger caspase-1 activation in macrophages should be identified.

4. In figure 2, it is better to choose caspase 1 mutant tumor cell line to perform caspase inhibitor experiments.

5. In figure 3g, whether caspase 1 KO PEM has the similar cell viability compared with WT PEM in coculture system or not. Because caspase 1 also plays an important role in cell death. It is better to add the results of CD36 and FABP4 expression level in caspase 1 KO TAMs to confirm the conclusion. Is tPPAR γ localized in nuclear in TAMs?

6. In figure 5, if tPPAR γ interacted with MCAD, it is better to analyze the location of tPPAR γ in TAMs.

7. In figure 7a-c, it is better to show the change of cell number and phenotype of TAMs in tumor-bearing mouse models by FCM analysis. The possibility whether the treatment with inhibitors directly impacts tumor cells should be excluded. Caspase1 $^{-/-}$ mice may be used for tumor growth.

8. Why knockdown of PPAR γ by shRNA failed to impact TAM differentiation?

Reviewer #3 (Remarks to the Author):

In this manuscript, the authors claim that Caspase-1 cleaves PPAR γ and that truncated PPAR γ is localized in mitochondria where PPAR γ and MCAD directly interact each other. The authors suggest that the modulation of MCAD expression affects TAM phenotype, leading them to conclude that the interaction of PPAR γ and MCAD attenuates MCAD activity and fatty acid oxidation.

Conceptually, since some caspase inhibitors are in clinical trials, it is important to know their mechanism of action. It is interesting that MCAD overexpression attenuates the tumor growth.

However, the actions of caspase inhibitors and MCAD overexpression may be explained without PPAR γ . Also, it is known that TAMs resemble M2 macrophages and PPAR γ is important for M2 polarization. But the TAMs in this manuscript are M1 macrophages. Given the accumulating evidence that high M1/M2 ratio exhibits anti-cancer property, much of the data shown in this manuscript is conflicting.

Experimentally, much of the data is not convincing and many experiments lack important controls. The other critical point that is missing here is to segregate the function of truncated PPAR γ from full length PPAR γ and PPAR γ that is not cleaved by caspase-1. For this, the relevant PPAR γ should be reconstituted in PPAR γ knockout cells.

Specific points:

1) Given the data that PPAR γ is not transcriptionally active in TAM, macrophages should decrease upon coculture. The authors should give an explanation for the discrepancy between Fig 4g and Fig 4n. Using PPAR γ knockdown and GW ligand is not enough to conclude that transcriptional activity is not the primary cause. The authors should examine this more directly by reconstituting PPAR γ that is lacking transcriptional activity in PPAR γ knockout cells.

2) The authors should show the effect of PPAR γ D64A and Δ 64D on gene expression as well as cellular localization, interaction with MCAD, MCAD activity and lipid content. As overexpression may affect endogenous PPAR γ , it should be done in PPAR γ knockout cells.

3) The authors claim that NLPR3 overexpression increases truncated PPAR γ in Supplementary Fig 2e, however, blot pattern looks different from other figures and those bands seem PPAR γ isoforms. If they are full length and truncated PPAR, the authors should run the gel longer. So as Fig 5d.

4) Fig 5e, what is plasma? Localization of PPAR γ in nuclei, mitochondria and cytoplasm as well as whole cell lysate by Western blotting and immunohistochemistry should be shown. Also, Western blot with fractionation should be run on the same gel to show the purity of fractionation.

5) In Fig 5a, why is the truncated PPAR γ size smaller than 40kDa?

6) Detection of octanoyl-CoA does not look sensitive enough. Standard should be shown.

7) What is the status of PPAR γ in MCAD overexpression?

Responses Letter

Major points:

1. Experimental repeats and statistics

We have added more description of experiments into the Methods (Experimental repeats and statistics). All immunoblotting assays had been performed at least three times to ensure the reproducibility of the experiments, and the representative pictures were shown in the final figures. TAM cells were isolated and pooled from at least two mice for every band in western blot or every sample in flow cytometry or RT-qPCR (e. g. in Fig. 1d, there are 8 bands, so at least 16 mice were used in Fig. 1d). All statistical analysis was generated based on at least three independent experiments.

2. in vivo studies using caspase-1^{-/-} mice

Based on the suggestion from reviewers, we have obtained *caspase-1^{-/-}* mice, and isolated BMDMs from *mutant* mice and control mice, and implanted these cells along with cancer cells in nude mice. These additional *in vivo* studies strongly supported our hypothesis that caspase-1 does play a critical role in TAMs differentiation, and therefore deficiency in caspase-1 impairs the TAM's ability of promoting tumor growth exactly. All new data are shown in Fig. 2 in this revised version. The results from *in vivo* studies using *caspase-1^{-/-}* mice make our conclusion more sturdily.

3. PPAR γ deficient THP-1 cells

Thanks to the suggestions from reviewer#2, we have established specific PPAR γ knockout THP-1 cells, in which we overexpressed truncated PPAR γ and mutant form of full length PPAR γ (D64A). In this knockout background, it's more clear to demonstrate the role of tPPAR γ without the interference of full length PPAR γ . And we repeated the experiments in TAMs differentiation, MCAD activity, metabolic reprogramming, tPPAR γ translocation in mitochondria, and etc. All new data are shown in Fig. 5e-h, Fig. 6e-g and Supplementary Fig. 10.

Reviewer #1:

It was difficult to tell how many times each experiment was performed. The methods claim that statistics were calculated based on at least 3 independent experiments - but for experiments without statistics (western blots, RNA, for example) it is not clear how many times the experiment was performed.

Response: Please refer to major point #1.

1. *Figure 1 shows that caspase 1 inhibition can block PPAR γ cleavage in TAMs, indicating that caspase 1 mediates PPAR γ cleavage in TAMs. Confirming the co-culture results with TAMs isolated from caspase 1 treated mice is nice.*

Response:

Thanks for this comment. We agree that the *in vivo* cleavage of PPAR γ is equally important. As the reviewer suggested, we have examined PPAR γ cleavage in isolated TAMs isolated from DMSO or caspase-1 inhibitor YVAD treated mice, and the results were shown in Fig. 1h. The results showed that indeed, the cleavage of endogenous PPAR γ is attenuated with caspase-1 inhibitor treatment, and confirmed the *in vitro* cleavage of PPAR γ .

2. *Figure 1c - Since the many following studies rely on the isolation of TAMs from the tumors please provide confirmation by flow or IF that the "sorted TAMs" are of high purity macrophages (>95% F480). Also, for Figure 1c, where did these cells come from 4T1 or MMTV?*

Response:

The purity of TAMs is a critical factor throughout our research. We have double checked the purity of TAMs in all experiments prior to downstream experiments. We have shown the morphological changes of cells before and after FACS sorting, and in this revision, we have included the quantification data of FACS sorting for TAMs, and the results are shown in Fig. S2a. A typical TAMs sorting will yield a purity of 95% with F4/80 staining.

In Fig. 1c, the sorted TAM cells were derived from MMTV-PyMT mice. Thank the reviewer for pointing this out, we apologize for the mistake and we have added this information in figure legends.

3. *There is no mention for how many times each western blot was run and how many mice were used in each experiment. Please provide details on sufficient number of repeats for western blots and TAM isolations in figure 1 (and throughout manuscript).*

Response:

Thanks for the comments. We regret our omission and have addressed this in the **Major points #1**. We have added more description of experiments into the Methods (Experimental repeats and statistics). All immunoblotting assays had been performed at least three times to ensure the reproducibility of the experiments, and the representative pictures were shown in the final figures. TAMs cells were isolated and pooled from at least two mice for every band in western blot or every sample in flowcytometry or RT-qPCR (e. g. in Fig. 1d, there are 8 bands, so at least 16 mice were used in Fig. 1d). All statistical analysis was generated based on at least three independent experiments.

4. *In supplementary Figure 2e the authors state "overexpression of HA-NLRP3 significantly increased the levels of the truncated form of PPAR γ and concomitantly decreased the expression of pro-caspase 1". Please provide (1) statistics and (2) a western blot with better separation, to see the tPPAR γ .*

Response:

We apologize for the low quality of western blotting. We have optimized the experimental protocol and provide better separation of tPPAR γ (Fig. 1j). We have provided the other 2 repeats in supplemental information (Supplementary Fig. 24).

5. *Figure 2 - the authors show that caspase 1 inhibition reverses PPAR γ cleavage and blocks TAM differentiation, as measured by RT-qPCR and cell based assays. While there is a statistically significant reduction in the "TAM phenotype" with caspase inhibitors, the phenotype is not entirely rescued. What accounts for the partial rescue?*

Response:

Our data suggested a critical role of caspase-1 in regulating PPAR γ cleavage that is required for TAMs differentiation. We are actually NOT expecting to see the full rescue of TAMs

phenotype by caspase-1 inhibition, because 1) pharmacological inhibitors are not able to inhibit the target completely. The degree of inhibition really depends on the mechanism that inhibitors bind to their targets (for example, reversible v.s. irreversible binding). 2) other mechanisms might partially contribute to the regulation of functional polarization of macrophage. For example, Lynda M Stuart group reported that phagocytosis of *S. aureus* by macrophage was regulated by NOX2 through ROS/inflammasome/caspase-1 pathway. However despite complete inhibition of ROS by DPI (an inhibitor of ROS), there remained a residual defect in control of phagosomes function in inflammasome-deficient cells. They indicated that other factors may be involved [1].

Indeed, in updated Fig.1g, we observed the remaining basal expression of truncated PPAR γ upon the caspase-1 inhibitor treatment, suggesting that the blockage of PPAR γ cleavage by caspase-1 inhibitor was NOT complete.

References:

[1]. Sokolovska, A. *et al.* Activation of caspase-1 by the NLRP3 inflammasome regulates the NADPH oxidase NOX2 to control phagosome function. *Nat Immunol.* 2013. 14: p. 543-553.

6. *In supplementary figure 4 - how many times were each of these experiments performed?
Taking 3 pictures of one well is not sufficient to generate statistics (S4a).*

Response:

Thank you for your nice suggestion. We regret our omissions in the previous manuscript and the omissions have been fixed in our revised manuscript. As we addressed in major points, all statistical analysis was performed based on at least three independent experiments. In this experiment, we calculated the mean value of 10 random fields taken under the microscope. Mean values of each independent experiments were used to perform the statistic analysis. We have added this information into figure legends of Supplementary Fig. 4 and Supplementary Fig. 5.

7. *Supplemental figure 5: IL-1b and IL-18 are found in the cell culture media of the co-culture system. The authors wanted to test if either of the caspase 1 substrates played a role in TAM differentiation. Adding exogenous IL-1b or IL-18 induces THP1 differentiation similar to MCF7 co-culture, but the authors conclude these cytokines are not involved in TAM differentiation. This is a confusing figure and conclusion.*

Response:

Thank you for the comments. It is known that IL-1 β and IL-18 are the major substrates of caspase-1 in the pro-inflammatory pathway [1]. In our research, we found that both IL-1 β and IL-18 were significantly upregulated in co-culture during TAMs maturation. Hence, we designed to add IL-1 β or IL-18 into *in vitro* coculture system to test if IL-1 β or IL-18 plays any role in TAMs differentiation. Surprisingly, there were no significant changes in TAMs differentiation, even on the higher cytokine concentration, which suggested that neither of above two factors promoted THP-1 differentiation (Supplementary Fig. 6).

References:

[1]. Afonina, I., Müller, C., Martin, S. & Beyaert, R. Proteolytic Processing of Interleukin-1 Family Cytokines: Variations on a Common Theme. *Immunity*. 2015. 42 (6): p. 991-1004.

8. *Please quantitate and provide statistics for 7J and 7L.*

Response:

Statistics for Fig. 9J and 9L (order after revision) have been provided in Supplementary Fig. 11b,c.

9. *It's interesting that the TAMs-MCAD are more phagocytic (7L) are there other markers that can show these macrophages are more "anti-tumor" like? Are they engulfing tumor cells? Do they increase co-stimulatory molecules (CD40 is shown in the microarray, was this confirmed at the protein level?). Again a decrease in "TAM differentiation" markers are shown in 7m, but is there an increase in anti-tumor markers?*

Response:

Thank you for your question. As you suggested, we have further confirmed that the phagocytic ability of TAMs-MCAD for tumor cells was also significantly elevated, and new data are shown in Supplementary Fig. 11d,e. We haven't observed the significant CD40 increase at the protein level in TAMs-MCAD. Actually, we would like to mention that CD40 pathway activation reveals a dual function for macrophages in cancer cells survival and invasion[1]. The exact role of CD40 expressed by macrophages remains elusive during the interaction between macrophages and tumors although the co-stimulatory action of CD40 has been proven in T cell [2].

Regarding the question “*are there other markers that can show these macrophages are more "anti-tumor" like?*” Up to now, there are very few reports about the biomarkers for TAM’s role in anti-tumoral effects except the phagocytosis enhancement of macrophages that has been well documented [3]. However, the pro-tumoral role of TAMs is well established, thus the biomarkers for such effects are available and tested extensively. Here, we have used a panel of well-known biomarkers for pro-tumoral role of TAMs not only, but also test the engulfing capacity of TAMs-MCAD with latex beads and tumor cells, convincingly demonstrated the elevated anti-tumoral function of TAMs.

References:

- [1]. Dumas, G. & Dufresne, M. CD40 pathway activation reveals dual function for macrophages in human endometrial cancer cell survival and invasion. *Cancer Immunol Immun.* 2013. 62: p. 273-283.
- [2]. Schoenberger, SP. *et al.* T-cell help for cytotoxic T lymphocytes is mediated by CD40-CD40L interactions. *Nature.* 1998. 393: p. 480-483.
- [3]. Willingham, SB. *et al.* The CD47-signal regulatory protein alpha (SIRPα) interaction is a therapeutic target for human solid tumors. *Proc. Natl. Acad. Sci. USA.* 2012. 109: p. 6662-6667.

10. *The authors should show that IN VIVO treatment of caspase 1 inhibitors decreases PPARγ cleavage and alters lipid accumulation in TAMs. I cannot see that this is ever done.*

Response:

IN VIVO treatment of caspase 1 inhibitors decreases PPAR γ cleavage was shown in Fig. 1h, alters lipid accumulation in TAMs was shown in Fig. 4g.

11. *A schematic would be helpful to the reader to put the whole story together.*

Response:

Thanks the reviewer for this comment. We’ve generated a colored schematic carton shown as the Fig. 10.

12. *What strain of MMTV-PyMT mice are used?*

Response: The mice strain is FVB/N-Tg (MMTV-PyVT) 634Mul/J

13. *This is beyond the scope of the work, but could the author speculate: What is the secreted MCF7-derived factor that induces the macrophage differentiation to a pro-tumor phenotype?*

Response:

This is an excellent question. Identification of factors that led to macrophage differentiation in the tumor microenvironment is one of hotspots in cancer research field, meanwhile very challenging. There are a lot of discussions focusing on the interaction between immune cells and tumor cells in recent years. However, there has been no virtual breakthrough yet except the milestone discovery of PD1/PDL1 checkpoint in tumor immunotherapy. We believe that the relative studies will be always the hot topics in the fields of tumor biology and cancer immunotherapy [1-4]. In our experiment, based on the results demonstrating that PPAR γ cleavage promoted TAMs differentiation, we found the increased expression of some key cytokines, such as IL-6, TNF α , IL-1 β , IL-18 in the co-culture system of MCF-7 and macrophages. Then each individual cytokine was added into macrophage medium to determine if the significant cleavage of PPAR γ can be observed. However, administration of above mentioned cytokines failed to induce PPAR γ cleavage and promote macrophage differentiation. The macrophages didn't acquire the ability to pro-invasion, pro-migration, and pro-angiogenesis and impaired phagocytic ability.

References

- [1]. Stroffnen, E. *et al.* Targeting of cancer neoantigens with donor-derived T cell receptor repertoires. *Science*. 2016. 352: p. 1337-1341.
- [2]. Kataoka, K. *et al.* Aberrant PD-L1 expression through 3'-UTR disruption in multiple cancers. *Nature*. 2016. 534: p. 402-406.
- [3]. Hugo, W. *et al.* Genomic and Transcriptomic Features of Response to Anti-PD-1 Therapy in Metastatic Melanoma. *Cell*. 2016. 165: p. 35-44.
- [4]. McGranahan, N. *et al.* Clonal neoantigens elicit T cell immunoreactivity and sensitivity to immune checkpoint blockade. *Science*. 2016. 351: p. 1463-1469.

Reviewer #2:

1. *The untreated THP-1 macrophages or the untreated macrophages isolated or derived from other sources were used control in the whole manuscript. The standardized M1 (LPS+IFN-g) and M2 (IL-4) macrophages should be used as controls as well, so that the authors can conclude the unique roles of caspase-PPAR γ in TAM differentiations. These are essential controls.*

Response:

Thanks for the comments. We felt in this context, standardized M1 and M2 macrophages are completely different cells as compared to TAM cells. TAM cells display a hybrid phenotype of

M1 and M2 cells. “*The unique roles of caspase-PPAR γ in TAM differentiations*” is due to the fact that cleavage of PPAR γ is not observed in either M1 or M2 cells (as the figure shows below), while we polarized the macrophages by LPS+IFN γ or IL-4 respectively. TAMs are differentiated from primary macrophages, with a mix population harboring both M1, M2 phenotypes, suggested a complicated mechanism during the differentiation. We are not using M1 or M2 as controls, simply because that TAMs is not differentiated or derived from either M1 or M2 cells. Therefore, in current research, we only compared the gene/protein profiles of TAMs to undifferentiated primary macrophages. M1 and M2 cells are only necessary if certain new markers need to be characterized. As all markers we used are gold-standard markers, it’s not necessary here to include M1 and/or M2 as controls.

No PPAR γ cleavage during the polarization of M1 or M2

(a,b) THP-1 macrophages were incubated for the indicated times with 20 ng/mL IFN γ and 100 ng/mL LPS, extracts of THP-1 cells were harvested at indicated time, PPAR γ and its truncated fragment in the polarization of M1 and M2 were determined by immunoblot analysis (a), NO production in the culture supernatant was measured (b). (c,d) Cells were incubated for the indicated times with 10 ng/mL IL-4, extracts of THP-1 cells were harvested at indicated time, PPAR γ and its truncated fragment in the polarization of M1 and M2 were determined by immunoblot analysis (c), relative arginase activity was measured (d).

2. *Specific inhibitors of caspase-3 and caspase-6 attenuate the cleavage of PPAR γ protein in response to TNF α in cultured adipocytes. The positive controls for inhibitors on caspase-3 and caspase-6 are needed in Fig. 1h and Fig. 1i.*

Response:

Thanks for the comments. The effects of caspase inhibitors are essential to the entire research. As reviewers suggested, we have included new data in supplementary Fig. 2g confirming the effectiveness of these inhibitors. However, in our research, we only detected the attenuation of the full length of PPAR γ upon the treatment of casp3/6 inhibitors, but not the cleavage fragment

of PPAR γ (Supplementary Fig. 2g). As we did not detect the increased activity of casp3/6 inhibitors in TAM-tumor co-culture system, so we did not exclude the possibility that caspase-3/6 could lead to cleavage of PPAR γ here.

3. *The soluble factors released by tumor cells to trigger caspase-1 activation in macrophages should be identified.*

Response:

Thanks for the comments. The activation of caspase-1 is a very complicated process, which could be in a cell autonomous and cell non-autonomous manner [1,2]. The soluble factors released by tumor cells could be various, e.g. multiple cytokines, chemokines, FFAs etc. [3,4]. To identify the exact soluble factors responsible for caspase-1 activation in TAMs differentiation need very intensive work. We have not go further to study the exact upstream mechanism that governs the caspase-1 activation in the current study, as it's beyond the scope of this study.

References:

- [1]. Lamkanfi, M. Emerging inflammasome effector mechanisms. *Nat Rev Immunol.* 2011. 11: p. 213-20.
- [2]. Rathinam, VA. & Fitzgerald, KA. Inflammasome Complexes: Emerging Mechanisms and Effector Functions. *Cell.* 2016. 165: p. 792-800.
- [3]. Kitamura, T., Qian, BZ. & Pollard, J. Immune cell promotion of metastasis. *Nat Rev Immunol.* 2015.15: p. 73-86
- [4]. Donath, MY. & Shoelson, SE. Type 2 diabetes as an inflammatory disease. *Nat Rev Immunol.* 2011.11: p. 98-107.

4. *In figure 2, it is better to choose caspase 1 mutant tumor cell line to perform caspase inhibitor experiments.*

Response:

Thanks for the suggestion. However, the goal of our current study is to establish the link between caspase-1 activity and pro-tumoral actions of TAMs. We focus on the caspase-1 performance in TAMs differentiation, not in tumor cells. We think the change in caspase-1 function induced by certain mutations in tumor cells is really unpredictable, which surely affect the interaction mode between tumor cells and macrophages. To elucidate the specific role of caspase-1 in TAMs differentiation and PPAR γ cleavage, we utilized caspase-1 knockdown THP-1 cells (Supplementary Fig. 5, new data), and BMDMs derived from *caspase-1*^{-/-} mice in coculture system (Fig. 2g), avoiding the side effects/off-target effects of caspase inhibitors. The results again confirmed the critical role of caspase-1 in TAMs differentiation.

5. In figure 3g, whether caspase 1 KO PEM has the similar cell viability compared with WT PEM in coculture system or not. Because caspase 1 also plays an important role in cell death. It is better to add the results of CD36 and FABP4 expression level in caspase 1 KO TAMs to confirm the conclusion. Is tPPAR γ localized in nuclear in TAMs?

Response:

Thanks for the comments. Caspase-1 KO PEM has the similar cell viability compared to WT PEM in coculture system, confirmed by MTT assay (as the figure shows below). Caspase-1 does play an important role in cell death; however, some researchers have also shown increased expression of caspase-1 does not always accompany with certain cell death due to the diversity in caspase-1 functionality. CD36 expression has been shown in Fig. 2g which reflects the comparable cell viability and also the effect of knockout of caspase-1.

Truncated PPAR γ is also found in nuclear. In supplementary Fig. 2b, the nuclear fraction was used in EMSA assay, as both PPAR γ and caspase-1 are present in nuclei.

Caspase-1 expression does not affect cell viability in coculture system.

Peritoneal macrophages were isolated from wild type (Co-Casp1-WT) or *caspase-1*^{-/-} mice (Co-Casp1-KO) and cocultured with 4T1. After 2 d, using wild type peritoneal macrophages cultured alone as control, viabilities of macrophages were determined by the MTT assay. There is no significant difference between experimental groups.

6. In figure 5, if tPPAR γ interacted with MCAD, it is better to analyze the location of tPPAR γ in TAMs.

Response:

Thank you. Our data in current research have shown that tPPAR γ localize in mitochondria (Fig. 6d&e, as well as in nuclei (Supplementary Fig. 2b)).

7. *In figure 7a-c, it is better to show the change of cell number and phenotype of TAMs in tumor-bearing mouse models by FCM analysis. The possibility whether the treatment with inhibitors directly impacts tumor cells should be excluded. Caspase1^{-/-} mice may be used for tumor growth.*

Response:

It's a great question. We have provided this data in Supplementary Fig. 11a, where we count the percentage of TAMs in total cells, and there is no difference between mice receiving different treatment.

We totally agree that genetic ablation mouse model is critical for current research. In this revision, we obtained *caspase-1^{-/-}* mice from Genetech[1]. In revised Fig. 2, we showed that implantation of EO771 cancer cells with wild-type BMDMs promotes the tumor growth, while caspase-1 deficient BMDMs significantly impaired this process. Corresponding changes have been made in revised manuscript.

References

[1]. Kuida, K. *et al.* Altered cytokine export and apoptosis in mice deficient in interleukin-1 β converting enzyme. *Science*. 1995. 267: p. 2000-2003.

8. *Why knockdown of PPAR γ by shRNA failed to impact TAM differentiation?*

Response:

Sorry for causing misunderstanding. Actually, knockdown of PPAR γ by shRNA failed to enhance TAMs differentiation in our coculture model compared with wild type THP-1 macrophages (Fig. 5a). Our research showed that tPPAR γ is critical and knockdown of PPAR γ also decrease the abundance of tPPAR γ , therefore treatment of sh-PPAR γ significantly impairs TAMs differentiation.

Reviewer #3:

Also, it is known that TAMs resemble M2 macrophages and PPAR γ is important for M2 polarization. But the TAMs in this manuscript are M1 macrophages. Given the accumulating

evidence that high M1/M2 ratio exhibits anti-cancer property, much of the data shown in this manuscript is conflicting.

Experimentally, much of the data is not convincing and many experiments lack important controls. The other critical point that is missing here is to segregate the function of truncated PPAR γ from full length PPAR γ and PPAR γ that is not cleaved by caspase-1. For this, the relevant PPAR γ should be reconstituted in PPAR γ knockout cells.

Response:

Thank you for your comments. It's been well accepted that TAMs in cancer tissues bear complicated characteristics rather than simply M2 or M1. And, it is noteworthy that TAMs exhibit non-classical M2/M1 phenotypes [1, 2]. Qian et al. convincingly showed a pro-tumoral role of TAMs, as enhanced levels of IL-6, TNF α and NF- κ B enable macrophages to promote tumor progression [1]. The detailed mechanism for TAMs differentiation remains a challenging task because the complexity of tumor microenvironment depending on tumor type and matrix constitution [2]. Our current research for the first time showed that wild type and caspase-1 deficient TAMs display significantly different ability to promote breast cancer cell growth, which is neither a typical M1 or M2. Importantly, the hallmarks for TAMs in our research have been well characterized by other researches: *IDO1/2* responsible for T cell inhibition[3]; *COX2*[4] and *VEGFA/C* are angiogenesis marker[5]; *MMP9* are used for pro-metastasis[6]. TAMs-hallmarks in mice such as *Arg1*, *Chil3 (Ym1)*[7], *Mmps*, *Ptgs2 (Cox2)*, *Cd274 (B7h1)*[8], *IDO1* and *Folr2 (Fr β)*[9]. All of the above genes are critical for pro-tumoral function associated with TAMs.

To segregate the function of truncated PPAR γ from full length PPAR γ and PPAR γ that is not cleaved by caspase-1, the relevant PPAR γ was reconstituted in PPAR γ knockout cells. Results are shown in the following specific points.

References

- [1]. Qian, BZ. & Pollard, JW. Macrophage Diversity Enhances Tumor Progression and Metastasis. *Cell*. 2010. 141(1): p.39-51.
- [2]. Noy, R. & Pollard, J. Tumor-Associated Macrophages: From Mechanisms to Therapy. *Immunity*. 2014. 41(1): p. 49-61.
- [3]. Zhao, QY. & Kuang, DM. Tumor-Associated Macrophages Privilege by Regulating IDO Expression in Activated CD69+ T Cells Foster Immune. *J Immunol*. 2012. 188: p.1117-1124.
- [4]. Ghosh, N. *et al*. COX-2 as a target for cancer chemotherapy. *Pharmacological reports: PR*. 2010. 62(2): p. 233.
- [5]. Schoppmann, SF. *et al*. VEGF-C expressing tumor-associated macrophages in lymph node positive breast cancer: Impact on lymphangiogenesis and survival. *Surgery*. 2006. 139: p. 839-46.
- [6]. Riabov, V. *et al*. Role of tumor associated macrophages in tumor angiogenesis and lymphangiogenesis. *Frontiers in Physiology*. 2014. 5.

- [7]. Yaddanapudi, K. & Putty, K. Control of Tumor-Associated Macrophage Alternative Activation by Macrophage Migration Inhibitory Factor. *J Immunol.* 2013. 190: p. 0-10.
- [8]. Bloch, O. *et al.* Gliomas Promote Immunosuppression through Induction of B7-H1 Expression in Tumor-Associated Macrophages. *Clinical Cancer Research.* 2013. 19(12): p. 3165-3175.
- [9]. Puig-Kröger, A. & Sierra-Filardi, E. Folate Receptor β Is Expressed by Tumor-Associated Macrophages and Constitutes a Marker for M2 Anti-inflammatory/Regulatory Macrophages. *Cancer Res.* 2009. 69(24): p. 9395-9403.

Specific points:

1. Given the data that PPAR γ is not transcriptionally active in TAM, macrophages should decreased upon the coculture. The authors should give an explanation for the discrepancy between Fig 4g and Fig 4n. Using PPAR γ knockdown and GW ligand is not enough to conclude that transcriptional activity is not the primary cause. The authors should examine this more directly by reconstituting PPAR γ that is lacking transcriptional activity in PPAR γ knockout cells.

Response:

It's very good point. However, we did not observe the alteration in cell number or cell viability in our coculture system upon the stimulation. The concentration of all caspase inhibitors used in our study was 50 μ M, which is the concentration widely used for inhibiting caspase activity, and the cytotoxicity was examined by MTT assay (as the figure shows below).

Caspase inhibitors do not alter cell viability

MCF-7 cells and THP-1 macrophages incubation with 50 μ M YVAD, 50 μ M VAD and 50 μ M or 100 μ M NCX-4016 respectively for 48 h, cell viabilities of both MCF-7 cells and THP-1 cells were determined with the MTT assay.

Regarding Fig. 4g and Fig. 4n, Fig 4g is *in vivo* experiment and Fig 4n is *in vitro* experiment.

Accumulation of lipids in TAMs *in vivo* might be due to increased synthesis of fatty acids in TAMs or result from increased lipid uptake from extracellular or other origins. However, in *in vitro* system, the sources of lipid are much less than *in vivo* system, lipid accumulation might be

simply due to the synthesis mechanism within TAM cells. Thank you very much for asking this questions, and we have added this description in **discussion** section.

We do agree that additional experiments in PPAR γ knockout cells are critical to acquire a convincing explanation. In this revision, we have performed the additional experiments and included new data in Fig. 5e-h. We established PPAR γ knockout THP-1 cells and re-expressed PPAR γ (D64A) mutant in this PPAR γ ^{-/-} THP-1 cells. We tested pro-tumor genes expression in coculture and PPAR γ activity (FABP4 expression), we can see both PPAR γ knockout and express PPAR γ (D64A) downregulated pro-tumor genes expression, however, PPAR γ activity is not impaired in PPAR γ (D64A) THP-1 cells (Fig. 5e). We also examined the metabolic state (Fig. 5f,g) to validate the functionality of PPAR γ . All these data shown that truncated PPAR γ contributes primarily to TAMs differentiation, but not full length PPAR γ activity.

2. *The authors should show the effect of PPAR γ D64A and Δ 64D on gene expression as well as cellular locazation, interaction with MCAD, MCAD activity and lipid content. As the overexpression may affect endogenous PPAR γ , it should be done in PPAR γ knockout cells.*

Response:

Thanks for the suggestions. We have included additional experiments demonstrating the effect of PPAR γ (D64A) and tPPAR γ on gene expression in (Fig. 5e, Supplementary Fig. 10), lipid content or lactate secretion (Fig 5h,i), MCAD activity (Fig. 6f,g). We have characterized the interaction between full length or truncated PPAR γ and MACD, and our data showed that only tPPAR γ strongly interacts with MCAD, but the interaction between the full length PPAR γ and MCAD is very weak (Fig. 6b&c). Mutation D64A prevents full length PPAR γ from truncating into small fragment, therefore, we did not test if mutated full length PPAR γ interacts with MCAD.

Meanwhile, we also showed here that the cleavage of PPAR γ would not affect the protein expression level of MCAD (Supplementary Fig.9g-i) in THP-1 cells and TAMs that were derived from mouse model; and the overexpression of MCAD would not have any effects on PPAR γ cleavage (Supplementary Fig. 9j).

These experiments certainly strengthened our hypothesis that only truncated form of PPAR γ plays an important role in promoting TAMs maturation, through interaction with MCAD. And the relationship is interaction-dependent, but not dose/protein expression level dependent.

3. *The authors claim that NLPR3 overexpression increases truncated PPAR γ in Supplementary Fig 2e, however blot pattern looks different from other figures and those bands seems PPAR γ isoforms. If they are full length and truncated PPAR, the authors should run the gel longer. So as Fig 5d.*

Response:

First we apologize for the low quality of this figure. We have optimized the experimental protocol and repeated the experiments with new blots showing in Fig. 1j. We have provided the other 2 repeats in supplemental information (Supplementary Fig. 24).

4. *Fig 5e, what is plasma? Localization of PPAR γ in nuclei, mitochondria and cytoplasm as well as whole cell lysate by Western blotting and immunohistochemistry should be shown. Also, Western blot with fractionation should be run on the same gel to show the purity of fractionation.*

Response:

Thanks for the comments. We apologize for any confusion. Plasma is lysate of the cytoplasmic proteins, from where we separated the nuclear proteins and mitochondria proteins. We've detected PPAR γ fragment expression in nuclei (as the figure shows below), and the nuclear fraction was used in EMSA assay, shown in supplementary Fig. 2b. We also repeated western blotting for cellular localization of tPPAR γ in mitochondria, and the proper controls are used for the same blot (Fig. 6d and e).

PPAR γ fragment also appears in nuclei.

Extracts of nuclear protein of THP-1 cells that cocultured with MCF-7 for 48 h in the absence or presence of YVAD were analyzed by immunoblot, LaminA/C as loading control. PPAR γ and tPPAR γ in whole cell lysates (WCL) in cocultured THP-1 macrophages was also shown.

5. *In Fig 5a, why the truncated PPAR γ size is smaller than 40kDa?*

Response:

This inconsistency is due to the technical issue. The gel post-silver-staining deformed and twisted a little which leads to the shift of protein ladder. We have repeated this experiments and

found the consistent results. We also stained the gel with coomassie blue staining, in which the gel kept intact and the band of tPPAR γ showed in 41kDa. We have included new gel figure in Supplemental Fig. 25.

6. *Detection of octanoyl-CoA does not look sensitive enough. Standard should be shown.*

Response:

Thanks for the suggestion. We have re-constructed the new cell lines (tPPAR γ was expressed in PPAR γ knockout THP-1 cells) and repeated the MCAD activity by measuring octanoyl-CoA. The standards have been determined in supplementary Fig.9e.

7. *What is the status of PPAR γ in MCAD overexpression?*

Response:

We have included new data showing MCAD overexpression or knockout does not alter the full length or truncated form of PPAR γ in coculture (Supplementary Fig. 9i), suggesting that MCAD does not affect the cleavage of PPAR γ .

Reviewer #4:

A. The authors convincingly demonstrate that a cleavage fragment PPAR γ plays a novel role in regulating the M2 macrophage phenotype in models of tumor induced macrophage polarization. They show that caspase-1 cleaves PPAR γ , resulting in a functional C-terminal PPAR γ fragment that translocates to the mitochondria, where it inhibits medium-chain acyl-CoA-dehydrogenase(MCAD), a regulator of fatty-acid chain oxidation. Inhibition of MCAD then promotes the M2 phenotype. Furthermore the authors show that inhibiting caspase-1 function or activating MCAD can repress tumor growth and the M2 phenotype of tumor associated macrophages. However, the authors do not explain how MCAD inhibition regulates the M2 phenotype.

Response:

Thanks for the question. We are particularly grateful for all the comments and invaluable suggestion by this reviewer, being supportive for our research. We have shown that inhibition of

MCAD by truncated PPAR significantly attenuated the metabolism regulation in mitochondria, which leads to the fatty acid oxidation dysfunction in general. Such cellular behaviors, considering as metabolic reprogramming, leads to the polarization changes of TAM. As suggested, to directly link the MCAD with TAM's differentiation, we have included new data here. In Figure 8, we show that MCAD is critical to TAM's differentiation, as the expression level of tPPAR γ had no effects on TAM's differentiation on the MCAD knockdown background. We believe the new data strengthened our conclusion on the critical of MCAD in TAMs's differentiation.

E. The conclusions are mostly appropriate but not as well developed as the body of work demands. The authors should discuss in greater detail how MCAD can promote the M2 phenotype, suggesting which transcription factors or epigenetic processes might be controlled directly by the change in fatty acid oxidation.

Response:

Thanks for the suggestion. We have included new contents in discussion explaining how MCAD is involved in promoting TAM's differentiation. Identifying key fatty acids or transcription factors that are regulated by MCAD and contribute to TAM's differentiation would be very important for understanding the mechanism in future, but such studies are beyond the scope of current research.

F. 1) I suggest that the authors work to improve the clarity of their writing. The authors have included a large body of experiments in this manuscript, and in some part of the manuscript, the description of the data or the conclusions drawn from the data are missing. To follow the experimental flow, the authors should emphasize both the data and the implications of the data for each experiment. In particular, the description of the effect of PPAR γ knockdown in gene expression is poor.

Respond:

Thanks for the suggestion. We have largely revised the text by English-native speaker. The description for PPAR γ knockdown experiments have been described in greater detail in method part.

2) *It is interesting that PPARgamma knockdown prevents M2 macrophage gene expression and that inhibition of PPARgamma cleavage does as well. Although the authors believe this is a discordant result, it is clear to me that PPARgamma expression is required so that its fragment can inhibit MCAD function. The authors should modify their discussion of this results to use it to support their mechanism.*

Respond:

Thanks for pointing out this. We have included this part in our discussion. Yes, as reviewer point out here, full length PPAR γ is the source of truncated PPAR γ , inhibiting full length PPAR γ will lead to the decrease in truncated PPAR γ , subsequently, leads to the disruption in TAM's differentiation.

3) *The authors should perform two studies to prove that PPARgamma effects on M2 macrophage polarization are due only to its inhibition of MCAD. They should perform a co-transfection with tPPARgamma and MCAD-OE to determine if the tPPARgamma can reverse the protective effect of MCAD-OE. In addition, PPARgamma shRNA knockdown in MCAD-KO cells could definitively show that PPARgamma's role in M2 polarization is due only to MCAD inhibition.*

Response:

Thanks for pointing out this. We have included this part in our newly revised manuscript, mainly as Figure 8. We determined the relative key gene expression in coculture cells with MCAD-OE, or MCAD-KO background, along with tPPAR γ -OE or KO. The data supports our conclusion that tPPAR γ regulates TAM's differentiation through MCAD. Please see revised manuscript and figures for detail.

4) *Throughout the manuscript, the authors refer to tumor associated macrophages (TAMs). However, they inappropriately use even TAMs when they should use the singular form. For example, in the second sentence of the manuscript, the authors write "Here we report that caspase-1 promotes TAMs differentiation..." In this case, the authors mean tumor associated macrophage (TAM) differentiation. Please correct the use of TAM versus TAMs throughout the manuscript.*

Response:

Thanks for the suggestions. We have checked all usage of TAM(s), and correct the mistakes in revised version of manuscript.

H. The manuscript is a bit technical. It would be helpful to focus on the big picture conclusions of each experiment.

Response:

Thanks for the suggestions. We have largely revised discussion part (highlighted in revised manuscript), and expanded the last part of discussion to elaborate an applicable picture of current research.

Table. List of revised figures.

Figure	Changes
Fig. 1g	New western blot replacing the old one
Fig. 1j	New western blot replacing the old one (supplementary data before revision)
Fig. 2a,g	CD36 was used as control, as suggested by reviewer#2
Fig. 2i-n	New experiments caspase-1 ^{-/-} mice were used for tumor growth as suggested by reviewer#2
Fig. 5e-h	TAMs differentiation, metabolism in PPAR γ knockout THP-1 or PPAR γ (D64A) reconstituted in PPAR γ knockout cells were shown, as suggested by reviewer#3
Fig. 5i,j	Position adjustment, Supplementary data before revision
Fig. 6d,e	tPPAR γ location in wild type THP-1 or PPAR γ (D64A) reconstituted in PPAR γ knockout cells, and the proper controls are used for the same blot to show the purity of fractionation, as suggested by reviewer#3
Fig. 6f,g	New experiments, we have re-constructed the new cell lines (tPPAR γ was expressed in PPAR γ knockout THP-1 cells) and repeated the MCAD activity by measuring octanoyl-CoA, as suggested by reviewer#3
Fig. 6h,i	Data of another repeat was added
Fig. 8	New figure demonstrates the indispensable role of MCAD, suggested by reviewer #4
Fig. 10	Schematic summary was added, as suggested by reviewer#1
Fig. S2a	Purity of TAMs was analyzed by flowcytometry as suggested by reviewer#1
Fig. S2b	Position adjustment, in Fig. 1 before adjustment
Fig. S2g	New experiments, the positive controls for inhibitors on caspase-3 and caspase-6 was shown as suggested by reviewer#2
Fig. S5	We utilized caspase-1 knockdown THP-1 cells to confirm the results from using caspase-1 inhibitors, as suggested by reviewer#2
Fig. S9f	New experiments, we showed that MCAD activity will not be affected in cell lines that are deficient in tPPAR γ , as suggested by reviewer#3
Fig. S9g	PPAR γ and PPAR γ fragment in coculture was added
Fig. S9i	New experiments, we showed that MCAD expression will not affect PPAR γ cleavage, as suggested by reviewer#3
Fig. S10	tPPAR γ was expressed in PPAR γ knockout THP-1 cells, TAMs differentiation was analyzed without the interference of full-length PPAR γ , as suggested by reviewer#3
Fig. S11a	We showed that in fig. 8a,b the tumor growth inhibition, was not due to TAMs content changes as suggested by reviewer#2
Fig. S11b,c	Statistics for 8J and 8L have been provided as suggested by reviewer#1
Fig. S11d,e	The ability of TAMs-MCAD engulfing tumor cells was shown as suggested by reviewer#1

Reviewers' comments:

Reviewer #1 (Remarks to the Author):

This is a resubmission to Nature Communication by Niu et. al. Niu et al demonstrate that PPAR γ is a novel substrate for Caspase 1. They show that Caspase1/PPAR γ /MCAD can be targeted therapeutically to block macrophages from becoming pro-tumor tumor macrophages. The authors provide extensive analysis of the mechanism and relationship of how caspase 1 can be inhibited to block TAM differentiation. They provide novel insight into macrophage biology, phenotype and metabolic activity. Importantly, they provide evidence that preventing TAM differentiation to a protumor phenotype reduces tumor burden in two different mouse models: the orthotopic 4T1 and transgenic MMTV model. Harnessing macrophages for cancer therapy (as opposed to depleting or blocking their function) is a very important and highly relevant in cancer immunotherapy right now.

One note on macrophage polarization – in agreement with the authors, it is difficult to assess tumor macrophage phenotype, as TAMs likely have phenotypes of both “M1” and “M2” macrophages. The authors included a short discussion on this (lines 47-51) will be helpful to readers who are not experts on macrophage biology/phenotype.

The authors have done an excellent job addressing my previous concerns and besides a couple of technical issues below this paper is a strong paper for publication at Nature Communication.

Some minor concerns:

1. There are still some grammatical issues (especially the use of TAMs vs TAM: line 53, 497, etc.)
2. Can the authors please provide a reference for lines 98-100.
3. Supl Fig 11a shows there is no change in the percent of macrophages between ctl treated mice and caspase inhibitor treated mice. What is this a percent of? Live cells? CD45+ cells?
4. In Fig. 9h, is there meant to be a difference in positive cells between the lacz and MCAD overexpressing cells? Can the authors please quantitate this? It appears there are more BMDM from the MCAD expression cells compared to the lacz control. Can you provide a brief explanation for why there is an increase in BMDM-MCAD cells? Especially since caspase inhibitor treatment does not change the percent of macrophages (Supl Fig 11a).
5. Lines 554-560 can you add one more line on how caspase 1 inhibition will be specific for macrophages?
6. Fig 1d – define “PM” in legend.
7. Supl Fig. 11 b and c should refer to Fig. 9 not 8.

Reviewer #2 (Remarks to the Author):

The authors have addressed some issues and concerns. However, addressing the following issues is still required:

- 1) There are many English program errors and typing errors cross the manuscript. It should be corrected.
- 2) It may be true that standardized M1 and M2 macrophages are extremely polarized macrophages and are different cells as compared to TAM cells. TAM cells may display a hybrid phenotype of M1 and M2 cells or a mix population of both M1 and M2 cells. If so, the gene/protein profiles of TAMs should be compared with undifferentiated primary macrophages, M1 and M2 cells. "The unique roles of caspase-PPAR γ in TAM differentiation" is due to the fact that cleavage of PPAR γ is not observed in either M1 or M2 cells. Thus, it is essential to show these data to readers to avoid missing leading. It also indicates that other cytokines or factors may trigger the activation of caspase-PPAR γ pathways during TAM differentiation.
- 3) It is important to show the tumor growth in caspase-1 $^{-/-}$ mice, but not co-injection of tumor cells with either WT or caspase-1 $^{-/-}$ macrophages.

Reviewer #3 (Remarks to the Author):

Although the authors have responded to some of the concerns raised, I continue to find the claim that cleaved PPAR γ moves to mitochondria to perform a completely different function to be far fetched and not adequately supported by data.

Reviewer #5 (Remarks to the Author):

This is a well-performed study revealing fascinating and interesting mechanism. However, it remains unclear how this immunometabolic regulation is achieved to control macrophage polarization and how TAM sustain their survival while continuously increase their intracellular lipid content.

Reviewer #6 (Remarks to the Author):

1. Niu et al demonstrate with appropriate controls and different approaches that PPAR γ is a new caspase1 substrate.
2. Regarding the mitochondrial translocation of tPPAR γ , data are less convincing. First, the authors should provide their readership the method used to isolate mitochondria in order to appreciate the putative purity of their extract. In addition, the authors should add further negative controls to support an efficient and pure mitochondria isolation. For instance, the authors should verify that specific markers from the ER, cytoplasm... are not expressed in the so-called mitochondria fraction; alternatively, the authors may show that VDAC1 is not present in other fractions. Finally, complementary experiments such as colocalisation assay as assessed by confocal or electron microscopy, would reinforce and support the current data.
3. The MCAD-mediated tPPAR γ effect on the metabolic reprogramming such as β -oxidation inhibition or oxygen consumption is not supported by current data. We may suggest that authors investigate the effect of tPPAR γ overexpression on respiration of TAMs for instance.

Response Letter

Reviewer #1

One note on macrophage polarization – in agreement with the authors, it is difficult to assess tumor macrophage phenotype, as TAMs likely have phenotypes of both “M1” and “M2” macrophages. The authors included a short discussion on this (lines 47-51) will be helpful to readers who are not experts on macrophage biology/phenotype.

Response: Thank you so much for your comments. As you suggested, we've included additional introduction regarding the mixed phenotypes of M1/2 for TAMs (in introduction). It indeed provided the insightful knowledge for the non-experts on macrophage biology.

1. There are still some grammatical issues (especially the use of TAMs vs TAM: line 53, 497, etc.)

Response: Thanks for pointing out. We have corrected all usage for TAMs vs TAM throughout the text.

2. Can the authors please provide a reference for lines 98-100.

Response: Thanks for pointing out. We have added one reference (Graham et al, 1996) for the statement of lines 98-100.

3. Supl Fig 11a shows there is no change in the percent of macrophages between ctl treated mice and caspase inhibitor treated mice. What is this a percent of? Live cells? CD45+ cells?

Response: This is the percentage of TAM cells in total primary cells that were extracted from MMT-PyMT tumors. The cells were labeled by F4/80 antibody. We've included the detailed information in figure legends.

4. In Fig. 9h, is there meant to be a difference in positive cells between the lacz and MCAD overexpressing cells? Can the authors please quantitate this? It appears there are more BMDM from the MCAD expression cells compared to the lacz control. Can you provide a brief explanation for why there is an increase in BMDM-MCAD cells? Especially since caspase inhibitor treatment does not change the percent of macrophages (Supl Fig 11a).

Response: Thanks for pointing out. In Fig. 9h, tumor bearing mice were intraperitoneally injected with MCAD-transduced BMDMs, or LacZ-transduced BMDMs. Because MCAD was HA-tagged, when we conduct immunofluorescence assay with anti-HA only MCAD-transduced BMDMs were detected; therefore, fluorescence intensity here represents MCAD expression, but not macrophage content. Also, according to our data, there is no change of macrophage content between these two groups (supplementary Fig. 12a).

5. Lines 554-560 can you add one more line on how caspase 1 inhibition will be specific for macrophages?

Response: In tumor microenvironment, the activated form of caspase 1 is only detected in

macrophages among all tumor tissue cells. Therefore, we believe here that the target of caspase-1 inhibitor is macrophages. We've included this explanation in revised manuscript (highlighted).

6. Fig 1d – define “PM” in legend.

Response: PM stands for peritoneal macrophage. We've included this in the legend for Fig 1d.

7. Supl Fig. 11 b and c should refer to Fig. 9 not 8.

Response: Thanks for pointing out. We've corrected this error.

Reviewer #2

1) There are many English program errors and typing errors cross the manuscript. It should be corrected.

Response: We've reviewed our manuscript carefully, and have a native English speaker reviewed and corrected for us.

2) It may be true that standardized M1 and M2 macrophages are extremely polarized macrophages and are different cells as compared to TAM cells. TAM cells may display a hybrid phenotype of M1 and M2 cells or a mix population of both M1 and M2 cells. If so, the gene/protein profiles of TAMs should be compared with undifferentiated primary macrophages, M1 and M2 cells. “The unique roles of caspase-PPAR γ in TAM differentiation” is due to the fact that cleavage of PPAR γ is not observed in either M1 or M2 cells. Thus, it is essential to show these data to readers to avoid missing leading. It also indicates that other cytokines or factors may trigger the activation of caspase-PPAR γ pathways during TAM differentiation.

Response: Thanks for the comments. We have performed microarray analysis and compared typical M1 and M2 markers in undifferentiated primary macrophages and TAMs, the gene profiles of TAMs display a hybrid phenotype of M1 and M2 cells. In addition, we found that YVAD which potently inhibited TAM markers has little effect on M1 and M2 marker gene profiles in TAMs (supplementary Fig.8e and f). Therefore, we conclude here that TAMs and M1, M2 macrophages share different differentiation mechanisms. Data demonstrating that cleavage of PPAR γ is NOT observed in either M1 or M2 cells were also shown (supplementary Fig. 8 a to d), which is also suggested by reviewer #2.

3) It is important to show the tumor growth in caspase-1^{-/-} mice, but not co-injection of tumor cells with either WT or caspase-1^{-/-} macrophages.

Response: Thank you so much for the suggestions. We totally agree that a genetic ablation mouse model is critical for current research. In this revision, we obtained caspase-1^{-/-} mice from Genetech. And we orthotopically implanted caspase-1^{-/-} mice or wild-type mice with EO771 cancer cells. we found that tumor growth in caspase-1^{-/-} mice was significantly suppressed, as compared to the tumor growth in wild type mice with same tumor cells. The results again

confirmed the critical role of caspase-1 in macrophage is supportive for tumor growth (revised Fig. 2 and revised Supplemental information P7, 8). Corresponding changes have been made in revised manuscript as well (revised manuscript P10).

Reviewer #3

Although the authors have responded to some of the concerns raised, I continue to find the claim that cleaved PPAR γ moves to mitochondria to perform a completely different function to be far fetched and not adequately supported by data.

Response: In this revision, we've completed a comprehensive studies on the translocation of tPPAR γ from nuclei to mitochondria. We've performed western blotting assay to demonstrate this translocation, and the purity of mitochondria fraction. In addition, we've shown the interaction between tPPAR γ between MCAD protein, which has been shown to be located in mitochondria exclusively. In this revision, by analyzing oxygen consumption rate (OCR), we further confirmed that tPPAR γ impairs mitochondria function; thus, our results convincingly demonstrated the important role of tPPAR γ in TAMs differentiation, which is mediated by alternating mitochondria metabolism.

Reviewer #5

This is a well-performed study revealing fascinating and interesting mechanism. However, it remains unclear how this immunometabolic regulation is achieved to control macrophage polarization and how TAM sustain their survival while continuously increase their intracellular lipid content. Too simple. Those down-stream studies are far beyond the scope of current study.

Response: Thanks for the comments. Indeed, the down-stream studies about the immunometabolic regulation is critical to understand the precise mechanism of which tPPAR γ determines the macrophage polarization by regulating the lipid metabolism. However, as the reviewer pointed out, these studies would be out of the scope of current investigation.

Reviewer #6

1). Niu et al demonstrate with appropriate controls and different approaches that PPAR γ is a new caspase1 substrate.

Response: Thanks for the comments. One of the novelties of current study is to provide the evidence that PPAR γ is indeed a substrate for caspase-1 in macrophage. The truncated form of PPAR γ is critical to promote the maturation of TAMs in tumor microenvironment.

2. Regarding the mitochondrial translocation of tPPAR γ , data are less convincing. First, the authors should provide their readership the method used to isolate mitochondria in order to appreciate the putative purity of their extract. In addition, the authors should add further negative controls to support an efficient and pure mitochondria isolation. For instance, the authors should verify that specific markers from the ER, cytoplasm... are not expressed in the so-called mitochondria fraction; alternatively, the authors may show that VDAC1 is not present in

other fractions. Finally, complementary experiments such as colocalisation assay as assessed by confocal or electron microscopy, would reinforce and support the current data.

Response: Thanks for pointing out the missing of this methods. We've included the detail information about the method of isolation of mitochondria. Briefly, we used a mitochondria isolation kit for cultured cells from Thermo Scientific (Cat# 89874). This kit has been widely used and well cited in many published literatures, such as Zhang et al., 2010 (*Nature*), Chen et al. 2009 (*PNAS*) and Rambold et al., 2011 (*PNAS*) (listed at the end of this letter). By using this kit, we have detected the presence of mitochondria marker, VDAC1 in these fractions. However, we failed to detect the presence of following markers for other organelles, GM130 (marker for Golgi), GRP78 (marker for ER), Na⁺,K⁺-ATPase (marker for membrane fraction) and β-actin (marker for cytoplasm), further verifying the purity of mitochondria fraction (supplementary figure 10c). We also tested the immunofluorescence staining for tPPAR_γ by confocal imaging; however, the antibody for PPAR_γ could not specifically recognize the protein. We've tested both antibodies from Santa Cruz (sc-7273 and sc-7196), neither of which are specific for immunostaining studies (data not shown). Finally, we've shown the specific interaction between tPPAR_γ and MCAD, whose expression was only detected in mitochondria. This further strengthened our hypothesis that tPPAR_γ is translocated from nuclei to mitochondria in a co-culture condition.

3. The MCAD-mediated tPPAR_γ effect on the metabolic reprogramming such as β-oxidation inhibition or oxygen consumption is not supported by current data. We may suggest that authors investigate the effect of tPPAR_γ overexpression on respiration of TAMs for instance.

Response: This is the great suggestion. To further explore whether tPPAR_γ affects mitochondria function through MCAD, we measured the oxygen consumption rates (OCRs) of THP-1 macrophages that overexpressing LacZ control or tPPAR_γ by Seahorse XF Analyzers (Agilent, Santa Clara, CA). Basal OCRs were measured, followed by serial additions of oligomycin (an inhibitor of ATP synthesis), carbonyl cyanide-p-trifluoromethoxyphenylhydrazone (FCCP; an uncoupling ionophore), and antimycin A (blocking agents for complexes I and III of the electron transport chain, respectively) to discern the relative contributions of mitochondrial and non-mitochondrial mechanism of oxygen consumption. We found that tPPAR_γ mediated β-oxidation inhibition reduced both basal OCRs and maximal respiratory capacity in mitochondria (Supplementary Fig. 11c). Moreover, Mitochondria content in each group was analyzed by mitotracker fluorescent probes, the result demonstrated that OCRs changes were not due to mitochondria content changes but mitochondria functions (Supplementary Fig. 11d).

Reference for mitochondrial isolation kit

Zhang, Qin, et al. "Circulating mitochondrial DAMPs cause inflammatory responses to injury." *Nature* 464.7285 (2010): 104-107.

Chen, Wen, et al. "Abcb10 physically interacts with mitoferrin-1 (Slc25a37) to enhance its stability and function in the erythroid mitochondria." *Proceedings of the National Academy of Sciences* 106.38 (2009): 16263-16268.

Rambold, Angelika S., et al. "Tubular network formation protects mitochondria from autophagosomal degradation during nutrient starvation." *Proceedings of the National Academy of Sciences* 108.25 (2011): 10190-10195.

REVIEWERS' COMMENTS:

Reviewer #2 (Remarks to the Author):

I agree with the authors' response with the additional data, so I like to accept it for publication.

Reviewer #6 (Remarks to the Author):

1. The authors should change « purity » by « enrichment » in the result section, since ER markers are still detected in the mitochondrial fraction (Sup Figure 10C).
2. Albeit not essential, colocalisation experiments have not been performed by confocal microscopy. Indeed, the authors argue for a lack of antibody specificity against PPAR γ . However they only tested 2 antibodies from Santa Cruz, while they could have used others from Abcam or Cell Signaling for instance.

REVIEWERS' COMMENTS:

Reviewer #2 (Remarks to the Author):

I agree with the authors' response with the additional data, so I like to accept it for publication.

Response: Thank you reviewer for the nice comment.

Reviewer #6 (Remarks to the Author):

1. The authors should change « purity » by « enrichment » in the result section, since ER markers are still detected in the mitochondrial fraction (Sup Figure 10C).

Response: Thanks for pointing out. We've corrected the word, and replaced with "enrichment". This indeed reflected the natural of such molecular translocation from nuclei to certain cellular organelles.

2. Albeit not essential, colocalisation experiments have not been performed by confocal microscopy. Indeed, the authors argue for a lack of antibody specificity against PPAR γ . However, they only tested 2 antibodies from Santa Cruz, while they could have used others from Abcam or Cell Signaling for instance.

Response: Thank you so much for pointing out the co-localization experiments. We've requested additional antibodies from Abcam (ab178860), and Cell Signaling Tech (#81B8) and tested in THP-1 cultured alone or cocultured with MCF-7 cells. In our hands, #81B8 from cell signaling tech did not show any detectable levels of PPAR γ in THP-1 cells by immunostaining experiments. While for both Santa Cruz and Abcam antibodies, we've observed the increased portion of PPAR γ in cytoplasm and the slightly translocation of PPAR γ to mitochondria when THP-1 cells were cocultured with MCF-7 cells (please see the attached figures in this letter). As both of antibodies recognize the full length of PPAR γ and the truncated form of PPAR γ , it's very difficult to identify whether the cytoplasmic expression of tPPAR γ is truncated form of PPAR γ or not only by immunostaining experiments. However, in figure 6d and 6e, we've convincingly showed the enrichment of tPPAR γ in mitochondria, by differentiated expression size of PPAR γ in immune blotting experiments. Therefore, we are requesting not to include the immunostaining figures.

Figure: The immunostaining for PPAR γ in THP-1 cells alone or cocultured with MCF-7 cells.

THP-1 cells were cultured on coverslips, alone or cocultured with MCF-7 cells for 48 hours. Standard protocol were applied here, and primary antibodies from santa cruz (sc-7273), Abcam (ab178860), and Cell Signaling Tech (#81B8) were used for PPAR γ , followed by FITC labeled specific secondary antibodies (green). Nuclei were stained with DAPI (blue), and Mitotracker (red) was used for mitochondria.

Anti-PPAR γ antibody from Santa Cruz (sc-7273)

Anti-PPAR γ antibody from abcam (ab178860)

Anti-PPAR γ antibody from cell signaling (#81B8)